# Position: Spectral GNNs Rely Less on Graph Fourier Basis than Conceived

**Yuhe Guo** [1]  **Huayi Tang** [1]  **Jiahong Ma** [1]  **Hongteng Xu** [1]  **Zhewei Wei** [1]

## Abstract

Spectral graph learning builds upon two foundations: Graph Fourier basis as its theoretical cornerstone, with polynomial approximation to enable practical implementation. While this framework has led to numerous successful designs, we argue that its effectiveness might stem from mechanisms different from its preconceived theoretical foundations.

In this paper, we identify two fundamental issues that challenge our current understanding: (1) The graph Fourier basis $\mathbf{U}$[1] faces too many questions to truly serve its intended role, particularly in preserving its semantic properties of Fourier analysis; (2) The limitations preventing expressive filters are not merely practical constraints, but fundamental barriers that naturally protect stability and generalization.

Importantly, the two issues entangle with each other. The second obscured the first: the natural avoidance of complex filters has prevented us from fully confronting the questions about $\mathbf{U}$'s role as a Fourier basis. This observation leads to our position: the effectiveness of spectral GNNs relies less on Graph Fourier basis than originally conceived, or, in other words, **spectral GNNs might not be so spectral**. The position leads us to at least two potential research interests: to incorporate a more semantically meaningful graph dictionary other than $\mathbf{U}$, and to re-examine the theoretical role of the introduced polynomial techniques.

## 1. Introduction

Spectral Graph Neural Networks (Spectral GNNs) represent a niche yet vibrant branch within the broader field of Graph

[1]Gaoling School of Articial Intelligence, Renmin University of China. Correspondence to: Zhewei Wei <zhewei@ruc.edu.cn>.

*Proceedings of the $42^{st}$ International Conference on Machine Learning*, Vancouver, Canada. PMLR 267, 2025. Copyright 2025 by the author(s).

[1]Eigenvectors of the normalized graph Laplacian.

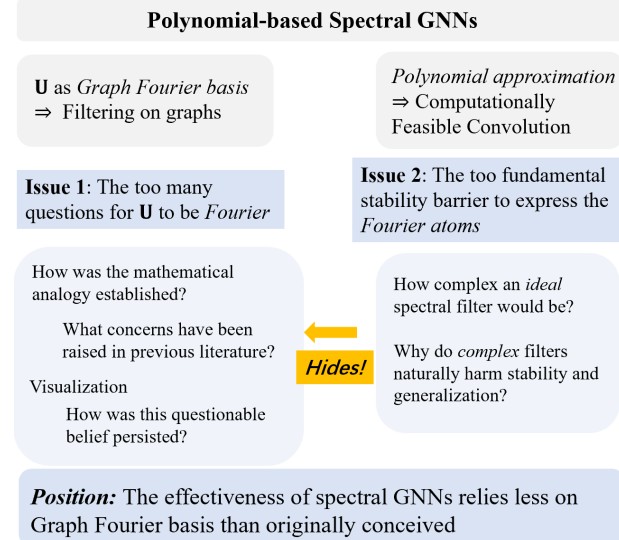

*Figure 1.* Organization of our position paper.

Neural Networks (GNNs). With lightweight parameters and simple architectures, Spectral GNNs have proven highly effective in node classification tasks. The appeal of this research branch stems from the underlying signal processing theory and polynomial fitting techniques. These methodologies and concepts not only hold a prominent place in current research trends but also have the potential to flow into advanced technologies like Kolmogorov–Arnold Networks (KAN) (Liu et al., 2024b; Bozorgasl & Chen, 2024; Liu et al., 2024a), and networks using Fourier features (Wang et al., 2021), thereby driving further innovation within these fields.

However, as spectral GNNs evolve and find applications in diverse scenarios, we identify two fundamental issues that challenge our understanding of their success (Figure 1):

- **The questionable but widely-accepted Fourier basis.** Do graph Laplacian eigenvectors truly serve as a Fourier basis? While the mathematical formalism can be systematically carried over, the semantic properties that make Fourier analysis powerful may not be preserved in this transition.

- **The inherent limitations in the filtering polynomial.** Why must we stay far from expressive filters despite

their theoretical appeal? The limitations preventing complex filters are not merely practical constraints, but fundamental barriers that naturally protect stability and generalization.

Most importantly, these two issues are not independent: **the second issue naturally obscures the first**. The inherent stability and generalization limitations of an 'expressive' filter have prevented us from fully confronting the questions about U's role as a Fourier basis.

In this paper, we identify and systematically examine these issues. For **issue 1**, we trace how the mathematical analogies were established, review concerns raised in previous literature, present experimental findings that question this analogy, and specifically, reflect on how this questionable belief has persisted. For **issue 2**, we demonstrate that complex filters inherently undermine stability and generalization. Our analysis considers both the stability of filter layers and GNN architectures, as well as the generalization behavior in two distinct scenarios. These findings lead to our position: **the effectiveness of spectral GNNs relies less on Graph Fourier basis than originally conceived.** This suggests that we need to re-examine the working mechanisms of spectral GNNs, potentially leading to (1) better understanding of how the polynomial techniques boost graph learning, and (2) designs that better align with a graph signal processing theoretical motivation.

## 2. Background

This section introduces the development of spectral graph learning. We want the reader to identify the two fundamental roots in Fig. 1 that form the basis of our position. We also introduce two basic graph learning settings, which are two scenarios in the sensitivity and generalization analysis of Section 4. Basic notations are introduced along the way, with a glossary table in Appendix A.

### 2.1. Background of Spectral Graph Learning

**Graph.** A graph $\mathcal{G} = (\mathcal{V}, \mathcal{E}, \mathcal{W})$ consists of a node set $\mathcal{V}$, an edge set $\mathcal{E}$, and edge weights $\mathcal{W}$. The connectivity pattern can be represented by an adjacency matrix $\mathbf{A}$, or its normalized version $\hat{\mathbf{P}}$. A vector $\mathbf{x} \in \mathbb{R}^n$ can be viewed as a graph signal, with $\mathbf{x} \to \hat{\mathbf{P}}\mathbf{x}$ represents a propagation of $\mathbf{x}$ on the graph. Another more important matrix we will use is the normalized Laplacian matrix $\hat{\mathbf{L}}$, whose eigen-decomposition plays a crucial role in spectral graph learning.

**Graph learning.** Graph Neural Networks (GNNs) can be categorized into three main approaches based on their design philosophy in utilizing the underlying structural information. In **spatial** domain models (Kipf & Welling, 2017; Hamilton et al., 2017; Veličković et al., 2017; Chen et al., 2020; Xu et al., 2018), graph structure serves as a medium for informa-

*Table 1.* Analogies between classical and graph domain concepts. A detailed version with concept *dependencies* and clear *explanations* of each notation is provided in Appendix B.

| Concept | Classical Domain | Graph Domain |
|---|---|---|
| Gradient | $\nabla : \mathcal{C}^1(\mathbb{R}^n)$ $\to (\mathcal{C}^0(\mathbb{R}^n))^n$ | $\nabla_{\mathcal{G}} : \mathbb{R}^{|\mathcal{V}|} \to \mathbb{R}^{|\mathcal{E}|}$ |
| Divergence | $\mathrm{div} : (\mathcal{C}^1(\mathbb{R}^n))^n$ $\to \mathcal{C}^0(\mathbb{R}^n)$ | $\mathrm{div}_{\mathcal{G}} : \mathbb{R}^{|\mathcal{E}|} \to \mathbb{R}^{|\mathcal{V}|}$ |
| Laplacian | $\Delta = \mathrm{div} \cdot \nabla$ | $\mathbf{L} = \mathbf{D} - \mathbf{A}$ |
| Laplacian Eigen Equation | $\Delta \phi = \lambda \phi$ | $\mathbf{L}\mathbf{v} = \lambda \mathbf{v}$ |
| Fourier Basis | $\{\mathrm{e}^{\mathrm{i}2\pi kx}\}_{k \in \mathbb{Z}}$ | $\{\mathbf{v}_k\}_{k=1}^{|\mathcal{V}|}$ |
| Fourier Transform | $\hat{f}(k) =$ $\int_{\mathbb{R}} f(x)\mathrm{e}^{-\mathrm{i}2\pi kx} \, dx$ | $\hat{f}(k) =$ $\sum_{i=1}^{|\mathcal{V}|} f(i)\mathbf{v}_k^*(i)$ |
| Inverse Fourier Transform | $f(x) = \sum_{k \in \mathbb{Z}}$ $\hat{f}(k)\mathrm{e}^{\mathrm{i}2\pi kx}$ | $f(i) =$ $\sum_{k=1}^{|\mathcal{V}|} \hat{f}(k)\mathbf{v}_k(i)$ |
| Convolution | $(f * h)(x) = \int_{\mathbb{R}} f(y)$ $h(x - y)\mathrm{d}y$ | $(f * h)_{\mathcal{G}} = \sum_{k=1}^{|\mathcal{V}|}$ $\hat{h}(\lambda_k)\hat{f}(k)\mathbf{v}_k$ |

tion propagation, where each layer intuitively corresponds to the process of neighbor features propagating to and influencing central nodes. **Matrix function** approaches view graphs as inputs to matrix functions (Maron et al., 2018; Keriven & Peyré, 2019), focusing on learning functions that are both invariant or equivariant to node ordering and expressive. **Spectral** approaches, which we focus on in this work, provide a more theoretical perspective by incorporating ideas from graph signal processing.

**Graph signal processing background.** Graph Signal Processing (GSP), rooted in spectral graph theory and classical signal processing took shape as a formal framework in the 2010s (Hammond et al., 2009; Shuman et al., 2013a; Sandryhaila & Moura, 2013; 2014) . It primarily aims to extend classical signal processing techniques to irregular graph-structured data, including filtering, compression, denoising, windowed Fourier transforms, and wavelets, among others.

In Table 1, we systematically summarize key analogies between classical and graph-based signal processing. At its core is the Graph Fourier Transform (GFT), which is built upon the eigenvectors of the normalized Laplacian matrix $\hat{\mathbf{L}} = \mathbf{I} - \hat{\mathbf{P}} = \mathbf{U}\boldsymbol{\Lambda}\mathbf{U}^\top$. The eigenvectors $\{\mathbf{u}_i\}_{i=1}^n$ serve as **atomic components** of the graph Fourier basis, playing the same role as the complex exponentials $\{\mathrm{e}^{\mathrm{i}2\pi kx}\}_{k \in \mathbb{Z}}$ in classical Fourier analysis.

**Basic spectral GNNs and spectral filtering.** Spectral

*Table 2.* From Graph Fourier Transform to Polynomial Filters.

| Operation | Matrix Form |
|---|---|
| Graph Fourier Transform | $\hat{\mathbf{x}} = \mathbf{U}^\top \mathbf{x}$ |
| Spectral Modulation | $\hat{\mathbf{x}}^* = \mathrm{diag}\{\theta_0, \theta_1, \ldots, \theta_{n-1}\}\hat{\mathbf{x}}$ |
| Graph Convolution | $\mathbf{x}^* = \mathbf{U}\,\mathrm{diag}\{\theta_0, \theta_1, \ldots, \theta_{n-1}\}\mathbf{U}^\top \mathbf{x}$ |
| Polynomial Approximation | $\theta_i \approx h(\lambda_i) = \sum_{k=0}^{K} \alpha_k g_k(\lambda_i)$ |
| Matrix Polynomial Form | $\mathbf{x}^* = \sum_{k=0}^{K} \alpha_k g_k(\hat{\mathbf{L}})\mathbf{x}$ |

GNNs was first introduced by Bruna et al. (2013) in image processing as a counterpart for spatial networks, and then extended to general graph domains (Henaff et al., 2015). To incorporate GSP into neural networks, *node features* $\mathbf{X} \in \mathbb{R}^{n \times d}$ are viewed as $d$ channels of *graph signals*. $\{\theta_i\}_{i=1}^n$ in table 1 (the third row) become learnable parameters, often with a small fraction of low-frequency eigenvectors preserved. Such a layer is called a **spectral filter**.

Notice that we move quickly through the spectral GNNs to provide readers with a high-level overview. For a more detailed treatment of this part, we refer readers to the background by Guo & Wei (2023b).

**Polynomial filters: the de facto formula for spectral GNNs.** While spectral filtering requires expensive eigen-decomposition, expressing filters as polynomials $h_\alpha(\lambda) = \sum_{k=0}^{K} \alpha_k g_k(\lambda)$ enables efficient computation: $h(\hat{\mathbf{L}})\mathbf{x} = \sum_{k=0}^{K} \alpha_k g_k(\hat{\mathbf{L}})\mathbf{x}$. Table 2 shows the step-by-step derivation from Graph Fourier Transform to polynomial filters, which forms the de facto formula for spectral GNNs. Note that as $K$ goes high, $g$ is expressive enough to approximate any $\{\theta_i\}_{i=1}^n$ under mild assumptions (Wang & Zhang, 2022).

Recent years have witnessed a surge of research focusing on enhancing the effectiveness of spectral GNNs through various polynomial techniques. These techniques include Chebyshev approximation (Defferrard et al., 2016; Kipf & Welling, 2017) and interpolation (He et al., 2022), Bernstein basis (He et al., 2021), Jacobi polynomial bases (Wang & Zhang, 2022), and Newton interpolation (Xu et al., 2024), among others. Additionally, advanced methods utilizing algebraic techniques such as the three-term recurrence relations and Krylov subspace methods (Guo & Wei, 2023b; Huang et al., 2024) have also been proposed. These spectral GNNs show state-of-the-art results on node classification tasks, and are capable of running on large graphs as Papers100M (Hu et al., 2020).

**Other spectral models.** A few spectral GNN variants lie outside the scope of Table 2. (i) Some approaches directly use decomposed eigenvectors or eigenbasis, targeting small graphs or settings with partial spectral informa-

tion (Lim et al., 2022; Geisler et al., 2024; Martirosyan et al., 2025). (ii) Some parameterize $h$ with (complex) rational functions (Bianchi et al., 2021; Levie et al., 2018; Li et al., 2025), instead of polynomials. (iii) Some works emphasize the combination of spectral and spatial mechanisms (Geisler et al., 2024; Guo et al., 2024; Guo & Wei, 2023a). These models fall outside our main analysis—we focus on filters of the form $h(\hat{\mathbf{L}})\mathbf{x}$, and Section 4 is devoted to that—but they illustrate how extensively the Graph Fourier basis is attempted to be utilized. Our analysis in the next section will question this usage.

### 2.2. Inductive and Transductive Graph Learning Tasks

In Appendix C, we introduce the inductive and transductive learning settings in graph learning, using graph classification and node classification tasks, respectively. Spectral filtering layers are integrated into the models. In Section 4, we will analyze how a 'complex' spectral polynomial filter can affect the stability and generalization of GNNs under inductive and transductive settings.

## 3. On the Validity of the Graph Fourier Basis

The first part of our position paper is to critically examine the validity of treating $\mathbf{U}$ as a graph Fourier basis.

We begin by pointing out the explicit or implicit assumptions held by spectral graph learning researchers that graph Fourier basis inherits similar semantics (*i.e.*, the global oscillatory patterns of different frequencies) as the continuous Fourier basis. After carefully examining the concepts' migration process between the continuous and discrete domains, we find that such an assumption is not self-evident and warrants careful scrutiny.

Then, we find that the concern has been raised by experts in graph signal processing, compressive sensing, and manifold, and present experimental evidence that visualizes these phenomena in empirical graphs.

Finally, as a position paper, we explore the origins of our unreliable assumptions with a historical perspective, identifying key sources that have shaped our understanding of graph Fourier bases. Through this comprehensive examination, we aim to uncover the gaps between formal derivations and the actual semantic properties of graph signals.

### 3.1. The Concern

Fourier bases were initially proposed for their invariance under the Laplacian operator, which proved advantageous for solving equations (O'Connor & Robertson, 1997). However, in spectral graph learning's narrative, as given in the background section, $\{\mathbf{u}_i\}_{i \in [1,n]}$ serve more as a dictionary of atoms (Shuman et al., 2013b; Zhang et al., 2012; Thanou

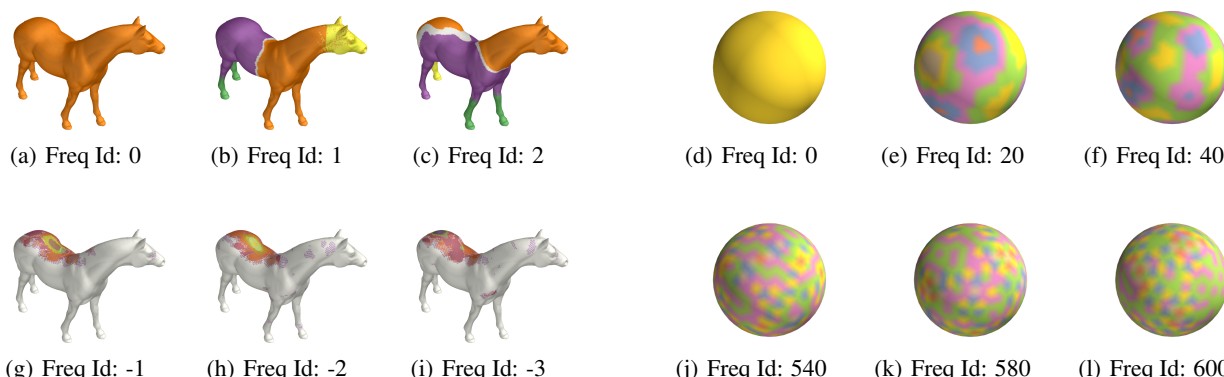

(a) Freq Id: 0    (b) Freq Id: 1    (c) Freq Id: 2    (d) Freq Id: 0    (e) Freq Id: 20    (f) Freq Id: 40

(g) Freq Id: -1    (h) Freq Id: -2    (i) Freq Id: -3    (j) Freq Id: 540    (k) Freq Id: 580    (l) Freq Id: 600

*Figure 2.* Visualization of low and high frequency eigenvectors on horse and sphere meshes. More plots are provided in Appendix E. **(1)** Colors represent the discretized values of eigenvectors, with the silver color representing the zero values. **(2)** *Top*: low-frequency eigenvectors; *Bottom*: high-frequency eigenvectors. *Left*: eigenvectors on horse mesh. *Right*: eigenvectors on sphere mesh. **(3)** For the horse mesh graph, low-frequency eigenvectors exhibit *smooth, global patterns* while the high-frequency signal bases, all with large silver regions, vanish on most of the vertices, exhibiting more *localized* patterns. **(4)** High-frequency eigenvectors exhibit more 'fourier' patterns on the 'regularized' sphere mesh graph than on the horse graph.

et al., 2014). It is natural to say, atoms that form a dictionary are expected to have "semantic" meanings (Bolinger, 1965; Ding et al., 2017).

Implicitly or explicitly, when adopting the graph Fourier basis, an assumption is that $\{\mathbf{u}_i\}_{i=1}^n$ possess *semantics* analogous to the continuous Fourier basis: each frequency component is expected to exhibit distinct **oscillatory patterns** that span the **entire domain**. This assumption is particularly evident when researchers are considering the heterophily degrees of graphs (Zhu et al., 2021; Luan et al., 2021; He et al., 2021; Luan et al., 2022; Zheng et al., 2022)—for heterophilic graphs, high-frequency atoms are regarded as crucial for fitting target signals that require *rapid oscillations* across edges, and it forms the motivation of most spectral graph learning works.

However, it is worth pausing to examine whether such a semantic assumption undoubtfully holds. In Appendix B, we systematically trace the analogies between concepts in the continuous domain and the graph domain. One can readily observe that, despite the systematic mathematical derivation being carried over, fundamental discrepancies exist. To name a few, **graph gradients**, which is defined as *differences* along discrete edges, is clearly different from the continuous gradients; and when migrating **divergence**, there are discrete numbers of *directions* defined upon edges. As these fundamental differences cascade, it is unrealistic to expect the semantic properties of continuous Fourier bases to be preserved in their graph counterparts.

This discrepancy likely originates from its borrowing from manifold definitions (Chung, 1997), where the graph Laplacian was derived from manifold studies. The distinction

between *general graphs* and those generated from manifolds may lead to inherent differences, a concern that has been emphatically highlighted by manifold experts (Belkin & Niyogi, 2008). [2]

### 3.2. Previous discussions

Though neglected in the context of recent spectral graph learning, the doubtful semantic role of graph Fourier basis has been discussed or noted in several works, and the were examined in several aspects.

**Localization.** Localization is the phenomenon that the energy of some graph Fourier basis vector concentrate a small subset of vertices, which is contrary to the expected *global* oscillatory patterns of Fourier basis. Shuman et al. (2013b) noted this phenomenon primarily due to McGraw & Menzinger (2008)'s study on the synchronizability of oscillator networks, and found that it is different from theoretical analysis on random settings. When generalizing windowed Fourier analysis to graphs, Shuman et al. (2013b) emphasized that "the existence of *localized eigenvectors* can limit the degree to which our intuition from classical time-frequency analysis extends to localized vertex-frequency analysis of signals on graphs".

**Compressibility.** In comressive sensing, Zhu & Rabbat (2012) highlighted the lack of theoretical analysis on when graph Laplacian eigenbases can be considered as Fourier transforms[3]. They assessed the "meaningfulness" of graph

---

[2]We will further explore this issue in the next section.

[3]Note: We quote their original statement here: "... none of the works applying graph Fourier basis provides a detailed theoretical analysis for when and why the graph Laplacian eigenbases can be

Fourier atoms through compressibility, which means that the signal can be represented efficiently using comparably fewer eigenvectors. They concluded that the meaningfulness of graph Fourier atoms is determined by two conditions: the target signal's total variance should be small on the graph, and the graph's eigenvalue distribution should be (generally) increasing.

**Concerns from manifold researchers.** Belkin & Niyogi (2008) cautioned that "viewing graphs as proxies for manifolds and applying manifold-inspired methods" has "few theoretical results" supporting its validity. Instead, how graph is generated from the underlying manifold is critical. This highlights that our generalization and utilization of Laplacian and its eigenvectors from manifolds to general graphs might lead to discrepancies. As cited by Zhang et al. (2012), Karni & Gotsman (2000) mentioned when compressing mesh data using graph Fourier basis, "the coordinates of a (sampled) vertex are very close to the average coordinates of its neighbors." This suggests that practitioners in graphics assume certain generation process of the underlying graph before applying graph Fourier bases, especially how the graph is sampled from a manifold.

### 3.3. Experimental Evidence

Following in discussion of Shuman et al. (2013b), we examine the localization phenomenon in empirical graphs.

**Visualization on meshes.** In our experiment, we explore the semantic properties of graph Fourier bases by visualizing eigenvectors at different frequencies (see Figure 2 and more plots in Appendix E). We selected horse[4] and sphere meshes as subjects, representing less-irregular and more-regular graph structures, respectively.

As shown in the figures, low-frequency eigenvectors on the horse mesh exhibit smooth, global patterns, while high-frequency eigenvectors concentrate on smaller subsets of vertices, displaying more localized patterns, which is consistent with the findings previously cited from Shuman et al. (2013b). It is also worth noting that, on the regularized sphere mesh, high-frequency eigenvectors exhibit more pronounced "Fourier" patterns, indicating that special highly-symmetric graph's structure significantly influences the behavior of eigenvectors.

**Measurement of localization on empirical graphs.** We follow McGraw & Menzinger (2008) to evaluate the localization of normalized and unnormalized eigenvectors on Cora dataset (Yang et al., 2016; Sen et al., 2008) by $\mathrm{loc}(\mathbf{u}_i) = \sum_{j \in [1,n]} u_{i,j}^4$. As shown in Figure 3, experi-

---

[4]Note: The horse mesh is downloaded from https://sites.cc.gatech.edu/projects/large_models/horse.html.

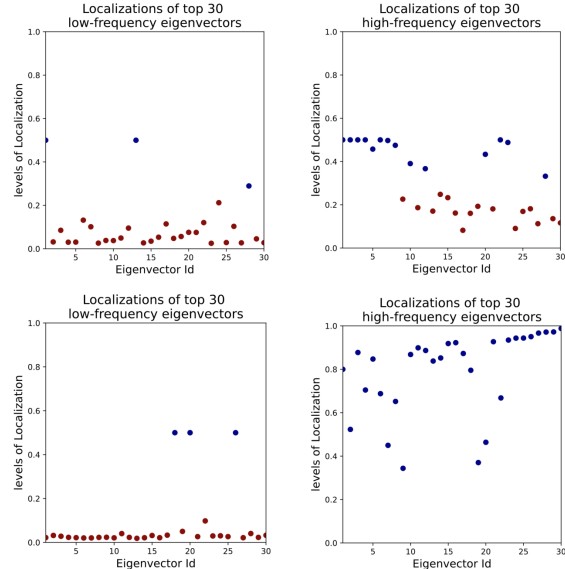

*Figure 3.* Localization of eigenvectors in Cora dataset. Blue color indicates highly-concentrated signal. **(1).** The **top row** represents eigenvectors from the normalized Laplacian, while the **bottom row** shows those from the unnormalized Laplacian. Within each row, the **left side** displays the first 30 eigenvectors (low-frequency), and the **right side** shows the last 30 eigenvectors (high-frequency). **(2)** High-frequency eigenvectors exhibit significant localization, which is partially alleviated by normalizing the Laplacian.

ments on Cora dataset exhibit **significant localization phenomenon** on high-frequency eigenvectors, which is partially alleviated by normalizing the Laplacian.

### 3.4. Origins of the Unreliable Assumptions

As a position paper, we want to take a step deeper to explore the origins of such an unreliable assumption, which has motivated a bunch of works, but remains questionable. In below, we discuss several sources that shape our **preconception** about graph Fourier bases.

Our misunderstanding primarily arises from two cognitive biases: (1) the tendency to **overgeneralize** well-established properties in *low-frequency* eigenvectors to *higher-frequency* ones, and to extend characteristics from *specific, well-studied* graphs to more general graph structures; and (2) the **significant influence** these "special" cases have on the research community. In the following sections, we will explore these biases further to uncover their implications on our assumptions about graph Fourier bases.

### Source 1: Well-understood and widely-used low-frequency eigenvectors.

The first source of our belief comes from the theoretically well-understood and widely-used properties of low-frequency Laplacian eigenvectors. These properties indi-

---

regarded as the Fourier transform of graphs."

cate that low-frequency eigenvectors can reflect the global community structure of the graph, as the global oscillation patterns of the first several Fourier basis. Such analogy in the low-frequency part might lead us to over-generalize this property to higher-frequency eigenvectors.

There are two well-known theorems that provide theoretical support for the soundness of using low-frequency eigenvectors to reflect the global community structure of the graph. **Cheeger's inequality** provides a lower bound of the second smallest eigenvalue of the Laplacian, and the eigenvector $\mathbf{u}_2$ can reflect the global community structure of the graph. The result can be extended to other low-frequency eigenvectors (Lee et al., 2014). **Nodal domain theorem** indicates that the number of nodal domains, which is the maximal connected subgraphs where $\mathbf{u}_k$ does not change sign, is at most $k$, and there exists at least one eigenvector corresponding to $\lambda_k$ that has exactly $k$ nodal domains. We put more concrete statements of these two theorems in Appendix F.

Von Luxburg (2007), from the perspective of relaxing the hard minimum cut problem, also derived the use of Laplacian eigenvectors in **spectral clustering**. Subsequently, convergence properties of spectral clustering based on specific graph generation assumptions were also provided (Tang & Priebe, 2018).

In practice, low-frequency eigenvectors are often used in various applications including graph drawing (Koren, 2003), and dimension reduction (Belkin & Niyogi, 2003), computer vision (Zhao et al., 2007), and even finance data analysis (Maslov, 2001). These applications make extensive use of the properties of low-frequency Laplacian eigenvectors, particularly their ability to reflect global structures and smooth transitions, which closely resemble the characteristics of low-frequency Fourier bases. This has led us to a tendency to overgeneralize these properties.

### Source 2: Special graphs with well-behaved (high-frequency) eigenvectors.

Another source that strongly reinforces our intuition about graph Fourier bases comes from the study of some **special graphs**, where *all* eigenvectors, including high-frequency ones, do possess clear semantic meanings and oscillatory patterns. This can be observed in our earlier visualization of eigenvectors on the sphere mesh, which exhibits more regular patterns due to its highly symmetric structure. Among these 'regular' graphs, ring graphs (cycle graph) and path graphs shape the most common intuition about graph Fourier bases, as Shuman et al. (2013b) pointed out. Their Laplacian eigenvectors are perfectly the Discrete Fourier Transform and Discrete Cosine Transform matrices (check Theorem.F.4 in Appendix F). Besides, for *grid graphs*, *Hamming graphs* and *hypercube graphs*, as the *Cartesian product* of special graphs, their properties are in some sense inherited

(Theorem.F.5). Specifically, the eigenvectors of hypercube graphs are known as *Hamming code*. Other examples include Johnson graph, and Cayley graph[5].

**Impact in fitness landscapes researches.** Why have these special graphs exerted such a profound influence on our understanding of graph Fourier bases? The profound influence of these special graphs, particularly their whole-spectra eigenvector properties, stems from a research interests in the study of **fitness landscapes** (Stadler, 1996; Reidys & Stadler, 2002), related to physics, engineering and optimization theory, where the solution spaces are often models as a special graph. For example, the landscape of a $p$-spin glass system (Binder & Young, 1986) is modeled as a hypercube graph $\mathcal{Q}_2^n$, where each *node* represents a possible spin configuration, and *edges* connect configurations that differ by one single spin flip. Another example is the landscape of the traveling salesman problem (TSP), where each *node* represents a possible tour, and *edges* connect tours that differ by a single edge swap. In these models, the eigenvectors of all frequencies exhibit clear 'energy' interpretations. The use of high-frequency Laplacian eigenvectors to analyze these landscapes has profoundly influenced our perception of high-frequency Laplacian eigenvectors in general graphs.

**Impact in computer graphics.** In addition, high-frequency Fourier bases have also been applied early in graphics (Taubin, 1995; Karni & Gotsman, 2000). They also noticed that fixed regular bases are beneficial for computation (Karni & Gotsman, 2001).

**Conclusion.** In summary, the combination of the aforementioned theoretically solid and widely influential special cases, along with our lack of careful consideration of the context of the theoretical derivations, together with the huge computational convenience of such Fourier bases when combined with polynomials, has led us to accept a concept that is not entirely solid.

## 4. On The Stability and Generalization Barriers to Expressive Spectral Filtering

In the following, we illustrate that, a complex and expressive enough polynomial that fully utilizes the graph Fourier bases would bring poor stability and generalization. As a result, we need to use "mild" filter polynomials, and the graph Fourier bases, mixed together, are hindered from being utilized as fundamental frequency signals.

---

[5]Note: The graphs mentioned here are note independent of each other. For example, the hypercube graph is a special case of Hamming graph, grid graph is a Cartesian product of path graphs, and Hamming graph $\mathcal{Q}_\alpha^n$ is an $n$-fold Cartesian product of complete graph $K_\alpha$.

### 4.1. The Concern

Let us consider what an "ideal" spectral filter would look like. Specifically, let $\mathbf{x} \in \mathbb{R}^n$ and $\mathbf{y} \in \mathbb{R}^n$ be the input signal and a target. For $i \in [n]$, the ideal response for the $i$-th simple eigenvalue $\lambda_i$ is $c_i = \mathbf{u}_i^\top \mathbf{y}/\mathbf{u}_i^\top \mathbf{x}$, where $\mathbf{u}_i$ is the eigenvector of the eigenvalue $\lambda_i$[6]. Theoretically, we seek a filter that can satisfy $h(\lambda_i) = c_i, \quad \forall i \in [n]$ (Wang & Zhang, 2022).

Such a filter would allow us to modulate each frequency component **independently**, similar to classical Fourier analysis. However, this ideal filter would lead to numerous issues. To see this, we visualize one 'ideal' filter in Figure 4 by plotting the values of ideal responses for each eigenvalue. Components with near-zero projections $\mathbf{u}_i^\top \mathbf{x}$ is excluded.

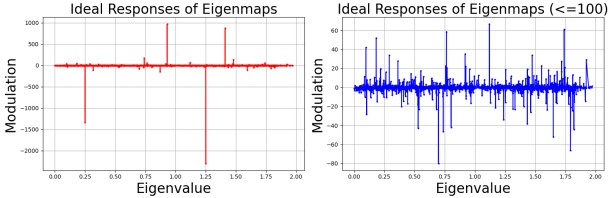

*Figure 4.* Visualization of an ideal spectral filter. *Left*: The original plot; *Right*: With values over 100 cut off. The graph is from Cora dataset. The original signal $x$ is derived by applying PCA for dimensionality reduction on the raw features, while the target $y$ is obtained by reducing the one-hot ground-truth labels.

As we have mentioned, an ideal filter should interpolate all these points $\{(\lambda_i, c_i)\}_{i=1}^n$. Thus, such a filer $h$ will face the following issues. (1) **Large maximum value of** $h$. It can be observed from Figure 4 that there exist several points with large response. Since $h$ should interpolate all these points, its maximum value $\max_{\lambda \in [0,2]} h(\lambda)$ is large. (2) **Large Lipschitz constant of** $h$. From Figure 4, we see that in order to correctly interpolate these points, the values of $h$ should change dramatically in some intervals (or $h$ is "sharp"). (3) **Large order would be needed** for $h$ to interpolate these points. In the next section, we demonstrate that both these three issues affect the stability of the filter $h$ and eventually hurt its generalization ability.

### 4.2. Stability of Polynomial Filter

Let us first analyze the **stability** of a polynomial filter $h$ by measuring the difference in its output $h_\alpha(\hat{\mathbf{L}})\mathbf{x}$ before and after perturbing the node features $\mathbf{x}$ and the normalized Laplacian matrix $\hat{\mathbf{L}}$. For simplicity, we stipulate that the polynomial bases are the monomial bases, that is, $h_\alpha(\lambda) = \sum_{k=0}^K \alpha_k \lambda^k$. Denote by $\mathbf{x}'$ and $\hat{\mathbf{L}}'$ be the node features and

---

[6] For multiple eigenvalues, the ideal response corresponds to how much should be scaled on the eigenspace. Although in experiments, we consider the case for multiple eigenvalues, we describe only the case of simple eigenvalues for simplicity.

the normalized Laplacian matrix after perturbing, our goal is to analyze the term $\|h_\alpha(\hat{\mathbf{L}})\mathbf{x} - h_\alpha(\hat{\mathbf{L}}')\mathbf{x}'\|$. Suppose that $\|\mathbf{x}'\| \le c_X$, we have

$$
\begin{aligned}
&\|h_\alpha(\hat{\mathbf{L}})\mathbf{x} - h_\alpha(\hat{\mathbf{L}}')\mathbf{x}'\| \\
=& \|h_\alpha(\hat{\mathbf{L}})\mathbf{x} - h_\alpha(\hat{\mathbf{L}})\mathbf{x}' + h_\alpha(\hat{\mathbf{L}})\mathbf{x}' - h_\alpha(\hat{\mathbf{L}}')\mathbf{x}'\| \\
\le& \|h_\alpha(\hat{\mathbf{L}})\|\|\mathbf{x} - \mathbf{x}'\| + \|h_\alpha(\hat{\mathbf{L}}) - h_\alpha(\hat{\mathbf{L}}')\|\|\mathbf{x}'\| \\
\le& \left( \sup_{\lambda \in [0,2]} h_\alpha(\lambda) \right)\|\mathbf{x} - \mathbf{x}'\| + c_X\|h_\alpha(\hat{\mathbf{L}}) - h_\alpha(\hat{\mathbf{L}}')\|.
\end{aligned}
$$

**Feature perturbation sensitivity and $h$'s largest response.** The above equation shows that the difference in output can be bounded by two terms. The first term composes the difference in *node features* before and after perturbations $\|\mathbf{x} - \mathbf{x}'\|$ and the maximum value of the graph filter $h$ over interval $[0, 2]$. The second term consists of the constant $c_X$ and the difference in *normalized Laplacian matrix* before and after perturbations $\|h_\alpha(\hat{\mathbf{L}}) - h_\alpha(\hat{\mathbf{L}}')\|$. For the first term, we conclude that the maximum value of the polynomial filter $h$ on $[0, 2]$ positively affects the stability of $h$. To guarantee that the polynomial filter $h$ is stable, we should restrict its range on $[0, 2]$.

**Graph structure perturbation sensitivity and $h$'s Lipschitz constant.** Now we analyze the term $\|h_\alpha(\hat{\mathbf{L}}) - h_\alpha(\hat{\mathbf{L}}')\|$ by using properties of the filter $h$. Before moving on, we introduce the definition of Lipschitz continuity.

**Definition 4.1** (Lipschitz Continuous). Let $h : [0, 2] \to \mathbb{R}$. We say $h$ is $C$-Lipschitz if $|h_\alpha(\lambda_1) - h_\alpha(\lambda_2)| \le C|\lambda_1 - \lambda_2|$ holds for any $\lambda_1, \lambda_2 \in [0, 2]$.

Notice that $C = \sup_{\lambda \in [0,2]} |h'(\lambda)|$ if $h$ is differential on $[0, 2]$. Therefore, the constant $C$ depicts the flatness of the curve $h_\alpha(\lambda) - y = 0$ with $\lambda \in [0, 2]$. Now we are in a place to see the connection between the Lipschitz constant $C$ and the the graph structure perturbation term $\|h_\alpha(\hat{\mathbf{L}}) - h_\alpha(\hat{\mathbf{L}}')\|$.

**Theorem 4.2** (Gama et al., 2020, Theorem 2). *Let $\hat{\mathbf{L}} = \mathbf{U}\mathbf{\Lambda}\mathbf{U}^\top$ be the eigenvalue decomposition of $\hat{\mathbf{L}}$. Define $\mathbf{E} = \hat{\mathbf{L}}' - \hat{\mathbf{L}}$ as the error matrix. Suppose that $\mathbf{E}$ is symmetric and thus its eigenvalue decomposition is given by $\mathbf{E} = \mathbf{V}\mathbf{B}\mathbf{V}^\top$. Define $\delta = (\|\mathbf{V} - \mathbf{U}\|_2 + 1)^2 - 1$. Suppose that $\|\mathbf{E}\| \le \varepsilon$ and $h$ is $C$-Lipschitz, then we have*

$$
\|h_\alpha(\hat{\mathbf{L}}') - h_\alpha(\hat{\mathbf{L}})\| \le C(1 + \delta\sqrt{n})\varepsilon + \mathcal{O}(\varepsilon^2). \quad (1)
$$

*Remark* 4.3. $\delta$ represents the eigenvectors misalignment between the normalized graph Laplacian matrix $\hat{\mathbf{L}}$ and the error matrix $\mathbf{E}$, as revealed in (Gama et al., 2020). Notice that there exists a difference between Theorem 4.2 and Theorem 1 in (Gama et al., 2020): they analyze the adjacent matrix perturbation $\|h_\alpha(\mathbf{A}) - h_\alpha(\mathbf{A}')\|$ while we analyze the normalized graph Laplacian matrix perturbation $\|h_\alpha(\hat{\mathbf{L}}') - h_\alpha(\hat{\mathbf{L}})\|$. Although the results are different, the proof process is the same as theirs.

Theorem 4.2 shows that the graph structure perturbation term $\|h_\alpha(\hat{\mathbf{L}}) - h_\alpha(\hat{\mathbf{L}}')\|$ can be further upper bounded by the Lipschitz constant $C$. A large value of $C$ implies a larger upper bound of the graph structure perturbation term and thus results in the polynomial filter $h$ being unstable, that is, applying perturbations on the graph structure may cause a drastic change in the output $h_\alpha(\hat{\mathbf{L}})\mathbf{x}$.

**Graph structure perturbation and $h$'s Order.** Notice that Theorem 4.2 requires the error matrix $\mathbf{E}$ to have a eigenvalue decomposition, which could not be satisfied under some scenarios. To this end, we adopt another technique introduced by Kenlay et al. (2020) to provide an upper bound for the graph structure perturbation term $\|h_\alpha(\hat{\mathbf{L}}) - h_\alpha(\hat{\mathbf{L}}')\|$. The derived result is as follows.

**Theorem 4.4** (Kenlay et al., 2020, Theorem 2). *Define $\mathbf{E} = \hat{\mathbf{L}}' - \hat{\mathbf{L}}$ as the error matrix, and $\tilde{\alpha} = (\alpha_1, \ldots, \alpha_K)$ as the vector of polynomial coefficients for all terms expect for the constant term. Suppose that $\|\mathbf{E}\| \leq \varepsilon$, for $K \geq 2$ we have*

$$\|h_\alpha(\hat{\mathbf{L}}') - h_\alpha(\hat{\mathbf{L}})\| \leq \frac{\|\tilde{\alpha}\|_1}{8}(K^2 - 1)\left(\frac{2(K+1)}{K-1}\right)^K \varepsilon.$$

*Remark* 4.5. The bound we present in Theorem 4.4 is slightly different from the one in Theorem 2 from (Kenlay et al., 2020). The reason of this difference is that the polynomial filter is defined as $h_\alpha(\hat{\mathbf{L}}) = \sum_{k=0}^{K} \alpha_k(\hat{\mathbf{L}} - \mathbf{I})^k$ in (Kenlay et al., 2020), while we define the polynomial filter as $h_\alpha(\hat{\mathbf{L}}) = \sum_{k=0}^{K} \alpha_k\hat{\mathbf{L}}^k$. Although the definitions are different, the proof process is the same as theirs.

Notice that Theorem 4.4 requires neither $h$ is $C$-Lipschitz continuous nor $\mathbf{E}$ is symmetric, which makes it cover more general cases than Theorem 4.2. Theorem 4.4 shows that the graph structure perturbation term $\|h_\alpha(\hat{\mathbf{L}}) - h_\alpha(\hat{\mathbf{L}}')\|$ can be upper bounded by the norm of polynomial coefficients and a function of $K$. Clearly, this function of $K$ grows as the increase of $K$, which indicates that a higher order of the polynomial filter $h$ lead to a larger upper bound of the graph structure perturbation term and thus making $h$ be unstable. Notice that a larger value of $K$ may also has a larger Lipschitz constant $C$. Thus, Theorem 4.2 and Theorem 4.4 convey the same insight that increasing the order $K$ results decreases the stability upper bound of $h$. Later we will see that a large value of $K$ may also result in the poor generalization of the spectral GNN with filter $h$.

### 4.3. Generalization of Spectral GNNs

In previous sections we have provided detail analysis on the stability of a *polynomial filter* $h$ by incorporating its property. Existing studies have shown that stability can be associated with a number of quantities that measure the performance of a learning model, such as generalization (Shalev-Shwartz et al., 2010) or robustness (Guo et al., 2023). Now, we

proceed on the *network generalization* and focus on two kinds of tasks: inductive graph classification and transductive node classification tasks, which are representative and widely adopted in various real-world application scenarios.

**Effects from magnitude of coefficients on inductive generalization.** To build connection between stability and generalization, we need to introduce a metric to measure the difference between two distinct graphs. In their seminal work, Chuang & Jegelka (2022) introduced a novel tree mover's distance (TMD) to measure the difference between two distinct graphs. Briefly, each node is represented by a depth-$L$ computation tree, TMD over two graphs is then the optimal transport cost matching two multisets of node-trees, and the tree-distance (TD) when calculating OT is defined recursively by solving an OT problem between their children down to the leaves. However, their definition can only assign identical weights to all neighbors in all layers. To remedy this issue, we propose the following modified tree mover's distance.

**Definition 4.6** (Modified Tree Distance). The modified tree distance between two trees $\mathcal{T}_a$ with root $r_a$ and $\mathcal{T}_b$ with root $r_b$ is defined as

$$\text{TD}(\mathcal{T}_a, \mathcal{T}_b) = \begin{cases} \text{OT}_{\text{TD}}(\rho(\mathcal{T}_{r_a}^{\#}, \mathcal{T}_{r_b}^{\#})) & \text{if } L > 1 \\ \left\|\frac{\mathbf{x}_{r_a}}{d^*(r_a)} - \frac{\mathbf{x}_{r_b}}{d^*(r_b)}\right\| & \text{otherwise} \end{cases}. \quad (2)$$

*Remark* 4.7. Our main modification lies in the definition of $d^*(u)$: $d^*(u) = 1$ if $u$ is the root of a tree $T$, and otherwise $d^*(u) = \deg(p(u)) \cdot d^*(p(u))$, where $p(u)$ is the parent of $u$ and $\deg(u)$ is the degree of $u$. Other notations align those in Definition 4 in (Chuang & Jegelka, 2022) and we refer readers to Appendix D for more detail. Different from Definition 4 in (Chuang & Jegelka, 2022), we do not need to compare the roots in Definition 4.6, since the adjacent matrix $\mathbf{A}$ dose not include self-loops in our definition.

Accordingly, we present a modified definition of tree mover's distance that is inherited from Definition 5 in (Chuang & Jegelka, 2022) (see Definition D.4 for more detail). Now we are in a place to analyze the Lipschiz constant of the following spectral GNN[7].

$$h_\alpha(\mathcal{G}, w) = \phi\left(\sum_{i=1}^{n}[h_\alpha(\mathbf{D}^{-1}\mathbf{A})\sigma(\sigma(\widetilde{\mathbf{X}}_a\mathbf{W}_1)\mathbf{W}_2)]_{i,:}\right).$$

**Theorem 4.8.** *Suppose that the activation function $\sigma(\cdot)$ is $K_\sigma$-Lipschitz and the graph readout function $\phi(\cdot)$ is $K_\phi$-*

---

[7]We use $\mathbf{D}^{-1}\mathbf{A}$ instead of $\hat{\mathbf{P}}$ or $\hat{\mathbf{L}}$ as the message passing matrix in this section. Note that $\mathbf{D}^{-1}\mathbf{A}$, known as random-walk Laplacian, has the same eigenvalues as $\hat{\mathbf{P}}$, and is also popular in applications (Shuman et al., 2013a).

*Lipschitz. Then we have*

$$\|h_\alpha(\mathcal{G}_a; w) - h_\alpha(\mathcal{G}_b; w)\|$$

$$\leq c_W^2 K_\phi K_\sigma^2 \left( \max_k |\alpha_k| \right) \left( \sum_{k=0}^K \text{TMD}^{k+1}(\mathcal{G}_a, \mathcal{G}_b) \right). \quad (3)$$

Next, we discuss how to use this result to establish a generalization bound for spectral GNN on the inductive node classification task. Following Chuang & Jegelka (2022), we consider the learning scenario that the distribution of training data points and test data points are different. Suppose that the data points in training are sampled from $\mu_S$. Let $\mu_T$ be the distribution of test data points and $\ell : \mathcal{W} \times \mathcal{Z} \to \mathbb{R}_+$, the expected risk on distribution $\mu_S$ and $\mu_T$ are defined as $R_S(w) = \mathbb{E}_{z \sim \mu_S}[\ell(w, z)]$ and $R_T = \mathbb{E}_{z \sim \mu_T}[\ell(w, z)]$, respectively. Now we are in a place to establish a upper bound for $R_T(w) - R_S(w)$ that depicts how well the model perform on unseen data points drawn from another distribution $\mu_T$. By combining Theorem D.5 and Theorem 1 in (Shen et al., 2018), we obtain the following result.

**Theorem 4.9** (Shen et al., 2018, Theorem 1). *For any spectral GNN $h(\cdot; w) \in \mathcal{H}_\mathcal{W}$, we have*

$$R_T(w) \leq R_S(w) + 2K\mathbb{W}_1(\mu_S, \mu_T) + \inf_{w' \in \mathcal{W}}[R_S(w') + R_T(w')], \quad (4)$$

*where $K$ is the Lipschitz constant of $h(\cdot; w)$ and $\mathbb{W}_1(\mu_S, \mu_T)$ is the domain discrepancy defined as*

$$\mathbb{W}_1(\mu_S, \mu_T)$$
$$= \inf_{\mu \in \Pi(\mu_S, \mu_T)} \int \sum_{k=0}^K \text{TMD}^{k+1}(\mathcal{G}_a, \mathcal{G}_b) d\mu(\mathcal{G}_a, \mathcal{G}_b).$$

By Eq. (3), the Lipschitz constant in Eq. (4) is $K = c_W^2 K_\phi K_\sigma^2 \max_k |\alpha_k|$. Therefore, we conclude that a small value of the magnitude of the polynomial coefficients make the spectral GNN be stable and have better out-of-domain generalization performance.

**Effects from order of polynomial and norm of coefficients on transductive generalization.** Now let us now turn to the transductive learning setting. In Theorem F.6, we shows that the transductive generalization gap $R_\text{test}(w, \pi) - R_\text{train}(w, \pi)$ can be upper bounded by two terms $L_1$ and $L_2$. Notice that the first term $L_1$ grows with the increase of $K$ since $\|\hat{\mathbf{L}}^k\| > 0$ for any $k = 0, \dots, K$. Therefore, using a higher order polynomial filter increase the term $L_1$. However, we could not conclude that increasing the order $K$ lead to a larger upper bound, as we are not sure whether the second term $L_2$ increases or decreases as $K$ increases. Besides, the second term reflect the effect of polynomial filters' coefficients on generalization. Notice

that $\|h_\alpha(\hat{\mathbf{L}})\|_\infty \leq \sum_{k=0}^K |\alpha|^k \|\hat{\mathbf{L}}^k\|_\infty$. If the absolute value of some coefficients are larger, the upper bound of $L_2$ will be large and thus implies that the spectral GNN may have a larger transductive generalization gap (Tang & Liu, 2023). Interestingly, these findings are the same as that obtained from Theorem 4.4, although the derivation of these two theorems are different. To summarize, the sufficient conditions for a polynomial filter $h_\alpha$ to be stable and have small transductive generalization gap is that both its order $K$ and the norm of its coefficients $\|\alpha\|_\infty$ are small.

### 4.4. Entanglement with the first issue

"There is a tradeoff between expressiveness and generalizability." This sounds a bit like a cliché, but in the context of spectral GNNs, it exhibits a particularly strong meaning. For example, since we have to avoid large Lipschitz constant, while spectral GNNs were designed to perform frequency-specific modulations, in practical polynomial filters, modulating one frequency component inevitably affects how its "neighbors" are modulated (since there are $|\mathcal{V}|$ eigenvalues in the range of $[0, 2]$). This behavior **fundamentally differs** from classical Fourier analysis, where frequency components can be manipulated independently. It hinders the ability of spectral GNNs to perform frequency-specific modulations, but from another angle, the natural avoidance of complex filters prevented us from fully confronting the questions about $\mathbf{U}$'s role as a Fourier basis.

## 5. Conclusion

In this position paper, we critically analyze the foundational assumptions of spectral graph neural networks, challenging the conventional understanding of spectral GNNs, thus providing new perspectives and directions for the future development of spectral GNNs.

## 6. Alternative Views

As introduced in the background and concern sections (Section 2 and 3.1), the whole development trajectory of spectral GNNs is primarily built upon the use of a dictionary known as the graph Fourier basis. As we have reviewed in Section 3.2, concerns about this dictionary have been discussed in other fields, but overlooked by practitioners of spectral GNNs. For this reason, we consider the default use of this dictionary as our alternative view.

## Acknowledgements

This research was supported in part by National Natural Science Foundation of China (No. 92470128, No. U2241212, No. 92270110), by Beijing Outstanding Young Scientist Program No.BJJWZYJH012019100020098, by Huawei-Renmin University joint program on Information Retrieval.

We also wish to acknowledge the support provided by the fund for building world-class universities (disciplines) of Renmin University of China, by Engineering Research Center of Next-Generation Intelligent Search and Recommendation, Ministry of Education, by Intelligent Social Governance Interdisciplinary Platform, Major Innovation & Planning Interdisciplinary Platform for the "Double-First Class" Initiative, Public Policy and Decision-making Research Lab, and Public Computing Cloud, Renmin University of China. The work was partially done at Beijing Key Laboratory of Research on Large Models and Intelligent Governance, MOE Key Lab of Data Engineering and Knowledge Engineering, Engineering Research Center of Next-Generation Intelligent Search and Recommendation, MOE, and Pazhou Laboratory (Huangpu), Guangzhou, Guangdong 510555, China.

## Impact Statements

This work advances machine learning methods for spectral graph learning and polynomial-based learning methods, with potential applications in practice. There are many potential societal consequences of our work, none of which we feel must be specifically highlighted here.

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

# A. Notations

Table 3. Summation of notations in this paper.

| Notation | Description |
|---|---|
| $\mathcal{G} = (\mathcal{V}, \mathcal{E}, \mathcal{W})$ | Undirected, connected graph with edge weights $\mathcal{W}$ |
| $n$ | Number of nodes in $\mathcal{G}$ |
| $\mathbf{D}$ | Degree matrix of $\mathcal{G}$ |
| $\mathbf{A}$ | Unnormalized adjacency matrix of $\mathcal{G}$ |
| $\hat{\mathbf{P}}$ | An symmetric-normalized adjacency matrix of $\mathcal{G}$, *i.e.*, $\hat{\mathbf{P}} = \mathbf{D}^{-1/2}\mathbf{A}\mathbf{D}^{-1/2}$ |
| $\mathbf{L}$ | Graph Laplacian matrix of $\mathcal{G}$, *i.e.*, $\mathbf{L} = \mathbf{D} - \mathbf{A}$ |
| $\hat{\mathbf{L}}$ | Normalized Laplacian matrix of $\mathcal{G}$, *i.e.*, $\hat{\mathbf{L}} = \mathbf{I} - \hat{\mathbf{P}}$ |
| $\mathbf{U}$ | Eigenvectors of $\hat{\mathbf{L}}$ and $\hat{\mathbf{P}}$ |
| $\lambda_i$ | The $i$-th eigenvalue of $\hat{\mathbf{L}}$, corresponding to $\mathbf{u}_i$ |
| $\mu_i$ | The $i$-th eigenvalue of $\hat{\mathbf{P}}$. $\mu_i = 1 - \lambda_i$ |
| $\mathbf{\Lambda}$ | Eigenvalues of $\hat{\mathbf{L}}$, *i.e.*, $\mathbf{\Lambda} = \text{diag}(\lambda_1, \lambda_2, \cdots, \lambda_n)$ |
| $\mathbf{x}$ | Input signal on one channel |
| $\mathbf{X} \in \mathbb{R}^{n \times d}$ | Input signals on $d$ channels |
| $\mathbf{Z} \in \mathbb{R}^{n \times d}$ | Filtered signals |
| $h(\cdot)$ | Filtering function |
| $h(\mathbf{L})\mathbf{x},$ | Filtering operation on signal $\mathbf{x}$. |
| $\{g_k(\cdot)\}_{k=0}^K$ | A polynomial basis of truncated order $K$ |
| $\{\alpha_k\}_{k=0}^K$ | Coefficients above a basis. i.e. $h(\lambda) \approx \sum_{k=0}^K \alpha_k g_k(\lambda)$ |
| $h_\alpha(\cdot)$ | Polynomial filter with coefficients $\alpha$ over the applied basis, *i.e.* $h_\alpha(\cdot) = \sum_{k=0}^K \alpha_k g_k(\cdot)$. |
| $\widetilde{\mathbf{X}} \in \mathbb{R}^{n \times \tilde{d}}$ | Original node features. |
| $\hat{\mathbf{L}}'$ | Laplacian of the perturbed matrix for sensitivity analysis |
| $\mathbf{x}'$ | Perturbed node feature (one dimension) for sensitivity analysis |
| $\mathbf{E}$ | Error matrix, i.e., $\hat{\mathbf{L}}' - \hat{\mathbf{L}}$ |
| $\varepsilon$ | Upper bound of error, i.e., $\|\mathbf{E}\| \leq \varepsilon$ |
| $\delta$ | Eigenvector misalignment, i.e., $(\|\mathbf{U} - \mathbf{V}\|_2 + 1)^2 - 1$ |
| $c_X$ | maximum norm of both vanilla and perturbed node features |
| $C$ | Lipschitz constant of filter $h$ |
| $m$ | Number of training nodes in transductive setting |
| $u$ | Number of test nodes in transductive setting |
| $w$ | Set of overall parameter of a model |
| $R_{\text{train}}(w, \pi)$ | Transductive training error of the model with parameter $w$ under partition $\pi$ |
| $R_{\text{test}}(w, \pi)$ | Transductive test error of the model parameter $w$ under partition $\pi$ |
| $\text{TMD}(\cdot, \cdot)$ | Modified Tree mover's distance of depth $L$ between two graphs |
| $\sigma$ | Activation function |
| $\phi$ | Graph readout function |
| $K_\sigma$ | Lipschitz constant of activation function $\sigma$ |
| $K_\phi$ | Lipschitz constant of graph readout function $\phi$ |
| $\mu_S$ | Distribution of the graphs for training in inductive learning |
| $\mu_T$ | Distribution of the graphs for test in inductive learning |

# B. Analogies

## B.1. Migration of Concepts to Graph Fourier Transformation

This appendix outlines the transition of graph Fourier transformation from classical signal processing.

Table 4 presents corresponding concepts between continuous and discrete domains, including gradient, divergence, Laplacian, and Fourier basis, leading to graph Fourier transformation and convolution in Table 5. Additional details are provided subsequently.

Notice that though we focus on unnormalized Laplacian here, the migration of concepts is also applicable to normalized Laplacian by simply weighting the two vertices of each edge in the graph gradient by their degrees.

*Table 4.* Concept Migrations From Gradient to Fourier Basis.

| No. | Prereq. | Classical Concept | Graph Domain Concept |
|---|---|---|---|
| (1) | - | Gradient $\nabla : \mathcal{C}^1(\mathbb{R}^n) \to (\mathcal{C}^0(\mathbb{R}^n))^n$ | Graph Gradient $\nabla_{\mathcal{G}} : \mathbb{R}^{|\mathcal{V}|} \to \mathbb{R}^{|\mathcal{E}|}$ |
| (2) | - | Divergence $\mathrm{div} : (\mathcal{C}^1(\mathbb{R}^n))^n \to \mathcal{C}^0(\mathbb{R}^n)$ | Graph Divergence $\mathrm{div}_{\mathcal{G}} : \mathbb{R}^{|\mathcal{E}|} \to \mathbb{R}^{|\mathcal{V}|}$ |
| (3) | (1,2) | Laplacian $\Delta = \mathrm{div} \cdot \nabla$ | Graph Laplacian $\mathbf{L} = \mathbf{D} - \mathbf{A}$ |
| (4) | (3) | Laplacian Eigen Equation $\Delta \phi = \lambda \phi$ | Graph Laplacian Eigen Equation $\mathbf{L}\mathbf{v} = \lambda \mathbf{v}$ |
| (5) | (4) | Fourier Basis $\{\mathrm{e}^{\mathrm{i}2\pi kx}\}_{k \in \mathbb{Z}}$ | Graph Fourier Basis $\{\mathbf{v}_k\}_{k=1}^{|\mathcal{V}|}$ |

*Table 5.* Concept Migration of Fourier Transforms and Convolutions.

| No. | Prereq. | Classical Concept | Graph Domain Concept |
|---|---|---|---|
| (6) | (5) | Fourier Transform $\hat{f}(k) = \int_{\mathbb{R}} f(x)\mathrm{e}^{-\mathrm{i}2\pi kx}dx$ | Graph Fourier Transform $\hat{f}(k) = \sum_{i=1}^{|\mathcal{V}|} f(i)\mathbf{v}_k^*(i)$ |
| (7) | (6) | Inverse Fourier Transform $f(x) = \sum_{k \in \mathbb{Z}} \hat{f}(k)\mathrm{e}^{\mathrm{i}2\pi kx}$ | Inverse Graph Fourier Transform $f(i) = \sum_{k=1}^{|\mathcal{V}|} \hat{f}(k)\mathbf{v}_k(i)$ |
| (8) | (6,7) | Convolution $(f * h)(x) = \int_{\mathbb{R}} f(y)h(x-y)dy$ | Graph Convolution $(f * h)_{\mathcal{G}} = \sum_{k=1}^{|\mathcal{V}|} \hat{h}(\lambda_k)\hat{f}(k)\mathbf{v}_k$ |

### B.2. Detailed Descriptions

(1). **Gradients**:

- Classical Gradient: $\nabla : \mathcal{C}^1(\mathbb{R}^n) \to (\mathcal{C}^0(\mathbb{R}^n))^n$, maps a *scalar field* to a *vector field*, measuring the change rates of the scalar field along each coordinate direction at every point. Here, $\mathcal{C}^1(\mathbb{R}^n)$ is the space of *once continuously differentiable* scalar fields on $\mathbb{R}^n$, and $(\mathcal{C}^0(\mathbb{R}^n))^n$ denotes $n$-dimensional vector fields with *continuous* components.
- Graph Gradient: $\nabla_{\mathcal{G}} : \mathbb{R}^{|\mathcal{V}|} \to \mathbb{R}^{|\mathcal{E}|}$, maps a signal on *vertex domain* to the *edge domain* by taking difference between the nodal domain signal values at the endpoints of each edge. For vertex $i$, the local gradient $[\nabla_{\mathcal{G}}\mathbf{f}]_i$ is defined as $\left[ \left\{ \frac{\partial \mathbf{f}}{\partial e}\big|_i \right\}_{e \in \mathcal{E} \text{ s.t. } e=(i,j) \text{ for some } j \in \mathcal{V}} \right]$, where for each edge $e = (i,j)$, $(\nabla_{\mathcal{G}}\mathbf{f})_{(i,j)} = \frac{\partial \mathbf{f}}{\partial e}\big|_i := f(j) - f(i)$

([Shuman et al.](), 2013a, Section II.E). The correspondence with the classical gradient arises because, in a graph, each edge plays the role of a coordinate direction in $\mathbb{R}^n$ along which partial derivatives are taken.

(2). **Divergence**:

- Classical Divergence: $\mathrm{div} : (\mathcal{C}^1(\mathbb{R}^n))^n \to \mathcal{C}^0(\mathbb{R}^n)$, maps a *vector field* to a *scalar field*. For a vector field $\mathbf{F}$, $\mathrm{div}(\mathbf{F}) = \sum_{i=1}^n \frac{\partial F_i}{\partial x_i}$, measuring how much the vector field spreads out (positive divergence) or converges (negative divergence) at every point.
- Graph Divergence: $\mathrm{div}_{\mathcal{G}} : \mathbb{R}^{|\mathcal{E}|} \to \mathbb{R}^{|\mathcal{V}|}$, maps a signal on *edge domain* to the *vertex domain*. For each vertex $i$, $\mathrm{div}_G(\mathbf{F})(i) = \sum_{j \in \mathcal{N}(i)} (F_{i,j} - F_{j,i})$, where $\mathbf{F} \in \mathbb{R}^{|\mathcal{E}|}$ is an edge signal.

(3). **Laplacian**:

- Classical Laplacian Operator: $\Delta : \mathcal{C}^2(\mathbb{R}^n) \to \mathcal{C}^0(\mathbb{R}^n)$, maps a twice-differentiable scalar field to a scalar field as the *divergence of the gradient*. The notation is $\Delta f := \mathrm{div}(\nabla f)$, and is also denoted by $\Delta f = \nabla \cdot \nabla f$.
- Graph Laplacian: $\mathbf{L}$. Migrating the definition of Laplacian operator as the divergence of the gradient, we have that the for a nodal domain signal $\mathbf{f}$ and a given a vertex $i$, $(\mathbf{Lf})_i$ should be $(\mathrm{div}_{\mathcal{G}} \nabla_{\mathcal{G}} \mathbf{f})_i$, where we first compute the gradient $(\nabla_{\mathcal{G}} \mathbf{f})_{(i,j)} = f(j) - f(i)$ and then $\mathrm{div}_{\mathcal{G}}(\nabla_{\mathcal{G}} \mathbf{f})(i) = \sum_{j \in \mathcal{N}(i)} ((\nabla_{\mathcal{G}} f)_{(i,j)} - (\nabla_{\mathcal{G}} f)_{(j,i)}) = \sum_{j \in \mathcal{N}(i)} ((f(j) - f(i)) - (f(i) - f(j))) = 2 \cdot \sum_{j \in \mathcal{N}(i)} (f(j) - f(i))$. Up to this point, when expressed in matrix form, the Laplacian operator should be written as $2\mathbf{A} - 2\mathbf{D}$. But ultimately, for scaling and sign convention, we use the definition $\mathbf{L} := -\frac{1}{2}(\mathrm{div}_{\mathcal{G}} \nabla_{\mathcal{G}} \mathbf{f}) = \mathbf{D} - \mathbf{A}$. We refer readers to ([Fu et al.](), 2022) for an illustrative example.

(4). **Eigenfunctions and Eigenvectors**:

- (Classical) Eigenfunction of Operator $T$: $\{\phi\}$, where $T\phi = \lambda\phi$ and $\phi \in \mathcal{C}^2(\mathbb{R}^n)$.
- (Graph) Eigenvector of Matrix $\mathbf{T}$: $\{\mathbf{v}\}$, where $\mathbf{Tv} = \lambda\mathbf{v}$ and $\mathbf{v} \in \mathbb{R}^{|\mathcal{V}|}$.

(5). **Fourier Basis**:

- Classical Fourier Basis: $\{e^{i2\pi kx}\}_{k \in \mathbb{Z}}$, solutions to the eigen equation of the Laplacian operator.
- Graph Fourier Basis: $\{\mathbf{v}_k\}_{k=1}^{|\mathcal{V}|}$, solutions to the eigen equation of the graph Laplacian.

## C. Graph Learning Settings: the Inductive and Transductive

We introduce the inductive and transductive learning settings in graph learning, using graph classification and node classification tasks, respectively. Spectral filtering layers are integrated into the models. We will analyze how a 'complex' spectral filter can affect the stability and generalization of polynomial spectral GNNs in Section 4.

**Inductive graph classification.** In this task, we are provided a set of data points $\{z_i\}_{i=1}^{n_{\mathcal{G}}}$, and each of them $z_i = (\mathcal{G}_i, y_i)$ is composed of a graph $\mathcal{G}_i = (\mathcal{V}_i, \mathcal{E}_i)$ and a unique associated label $y_i$. Our goal is to train a GNN model with these data points, which is able to predict the label of a new coming data point. Notice that this is an inductive task since the test data points are unseen during training. For the graph $\mathcal{G} = (\mathcal{V}, \mathcal{E})$ from each data point $z$, each node $v \in \mathcal{V}$ contains a $\tilde{d}$-dimensional feature and we denote it as $\widetilde{\mathbf{x}} \in \mathbb{R}^{\tilde{d}}$. The input features of the graph $\mathcal{G}$ is then defined as $\widetilde{\mathbf{X}} = [\widetilde{\mathbf{x}}_1^\top; \dots; \widetilde{\mathbf{x}}_n^\top] \in \mathbb{R}^{n \times \tilde{d}}$, where $n = |\mathcal{V}|$ is the total number of nodes in $\mathcal{G}$. Let $\phi : \mathbb{R}^{\tilde{d}} \to \mathbb{R}$ and $\sigma(\cdot)$ be the graph readout function and activation function, respectively. For a spectral GNN polynomial with filter $h_\alpha$, using $\mathcal{G}$ as input, its output is given by

$$h(\mathcal{G}; w) = \phi\bigg( \sum_{i=1}^n [h_\alpha(\hat{\mathbf{L}})\sigma(\sigma(\widetilde{\mathbf{X}}\mathbf{W}_1)\mathbf{W}_2)]_{i,:} \bigg), \tag{5}$$

where $\mathbf{W}_1, \mathbf{W}_2 \in \mathbb{R}^{\tilde{d} \times \tilde{d}}$ are learnable weight matrices. $\mathbf{A} \in \{0,1\}^{n \times n}$ and $\mathbf{D} \in \mathbb{R}^{n \times n}$ are the adjacent matrix and degree matrix of graph $\mathcal{G}$, respectively. We use $w = [\mathrm{vec}[\mathbf{W}_1]; \mathrm{vec}[\mathbf{W}_2]; \alpha_0; \dots; \alpha_K]$ to denote the collection of all learnable parameters. Let $\mathcal{W}$ be the space of parameter, then the hypothesis space can be represented as $\mathcal{H}_{\mathcal{W}} = \{h(\cdot; w) : w \in \mathcal{W}\}$.

The message passing matrix in (5) is sometimes given in other ways. Specifically, we use $\mathbf{D}^{-1}\mathbf{A}$ in theoretical analysis. This is because $\mathbf{D}^{-1}\mathbf{A}$ has the same eigenvalues as $\hat{\mathbf{P}} \equiv \mathbf{I} - \hat{\mathbf{L}}$ and is also popular in applications ([Shuman et al.](), 2013a). Then the spectral GNN is given by

$$h(\mathcal{G}; w) = \phi\bigg( \sum_{i=1}^n [h_\alpha(\mathbf{D}^{-1}\mathbf{A})\sigma(\sigma(\widetilde{\mathbf{X}}\mathbf{W}_1)\mathbf{W}_2)]_{i,:} \bigg). \tag{6}$$

**Transductive node classification.** In this task, we are provided **a fixed graph** where only partial nodes are labeled. Our goal is to train a GNN model with this graph to make prediction for those unlabeled ones. Concretely, denote by $\mathcal{G} = (\mathcal{V}, \mathcal{E})$ the fixed graph. Each node $v \in \mathcal{V}$ comprises features $\widetilde{\mathbf{x}} \in \mathbb{R}^d$ and an unique associated label $y \in \mathcal{Y}$, which is treated as a data point $z = (\widetilde{\mathbf{x}}, y)$. Then, the input features of this graph is represented as $\widetilde{\mathbf{X}} = [\widetilde{\mathbf{x}}_1^\top; \ldots; \widetilde{\mathbf{x}}_n^\top]$, where $n = |\mathcal{V}|$ is the total number of nodes in $\mathcal{G}$. Let $m$ and $u := n - m$ be the number of training and test nodes, respectively. The training and test in this learning task is generated by a random partition over the full nodes. Specifically, by continuously sample without replacement from $\{1, \ldots, n\}$ and place the obtained elements into a sequence $\pi = (\pi_1, \ldots, \pi_n)$, the training and test set are denoted as $\{z_{\pi_i}\}_{i=1}^m$ and $\{z_{\pi_i}\}_{i=m+1}^n$, respectively. After that, we feed all node features $\{\widetilde{\mathbf{x}}_i\}_{i=1}^n$, the graph structure $\mathcal{E}$, and labels of nodes in training set $\{y_{\pi_i}\}_{i=1}^m$ to the model, and requires it to predict the labels of the rest nodes. Notice that this is a transductive task since the features of test data points are visible during training. Let $\sigma(\cdot)$ be the activation function. For a spectral GNN polynomial with filter $h_\alpha$, using $\mathcal{G}$ as input, we denote its output by

$$h(\mathcal{G}; w) = \text{Softmax}(h_\alpha(\hat{\mathbf{L}}) \sigma(\sigma(\widetilde{\mathbf{X}} \mathbf{W}_1) \mathbf{W}_2)), \tag{7}$$

where $\mathbf{W}_1 \in \mathbb{R}^{\tilde{d} \times d'}, \mathbf{W}_2 \in \mathbb{R}^{d' \times |\mathcal{Y}|}$ are learnable weight matrices. Here, $w = [\text{vec}[\mathbf{W}_1]; \text{vec}[\mathbf{W}_2]; \alpha_0; \ldots; \alpha_K]$ is the collection of all learnable parameters. Let $\mathcal{W}$ be parameter space and $\ell : \mathcal{W} \times (\mathbb{R}^{\tilde{d}} \times \mathcal{Y}) \to \mathbb{R}_+$ be the loss function, the transductive training and test error are accordingly defined as $R_{\text{train}}(w, \pi) = \frac{1}{m} \sum_{i=1}^m \ell(w, z_{\pi_i})$ and $R_{\text{test}}(w, \pi) = \frac{1}{u} \sum_{i=m+1}^n \ell(w, z_{\pi_i})$, respectively.

# D. Proof

We first analyze the stability of this spectral GNN by measuring the difference in its output $\hat{y}$ of two different input graphs $\mathcal{G}$ and $\mathcal{G}'$. To this end, we need to define a specific metric to measure the discrepancy between two graph $\mathcal{G}$ and $\mathcal{G}'$. Chuang & Jegelka (2022) recently introduce a novel definition of such metric named "tree move distance" and demonstrate that it can well describe the stability and generalization of GNN. To introduce the definition of tree move distance, we need two prerequisite concepts: computational tree and blank tree augmentation.

**Definition D.1** (Chuang & Jegelka, 2022, Definition 1). Let $\mathcal{G} = (\mathcal{V}, \mathcal{E})$ be a given graph. For each node $v \in \mathcal{V}$, denote by $T_v^L$ the depth-$L$ computation tree of $v$, which is defined as follows: (1) $T_v^1 = v$; (2) for $2 \leq l \leq L$, for each node $\tilde{v}$ at the $(l-1)$-th level of $T_v^L$, find all neighbor nodes $\mathcal{N}(\tilde{v})$ of $\tilde{v}$ in the graph $\mathcal{G}$ and connect them with $\tilde{v}$ as its leaf nodes. The multiset of depth-$L$ computation trees induced by $\mathcal{G}$ is defined as $\mathcal{T}_{\mathcal{G}}^{\#, L} = \{\{T_v^L\}\}_{v \in \mathcal{V}}$.

**Definition D.2** (Chuang & Jegelka, 2022, Definition 2 and Definition 3). Define $T_{\mathbb{0}}$ as a blank tree that contains only the root node, whose feature is the zero vector $\mathbb{0}$. Then, $T_{\mathbb{0}}^n$ is the multiset containing $n$ blank trees. For two multisets of trees $\mathcal{T}_{\mathcal{G}}^\#$ and $\mathcal{T}_{\mathcal{G}'}^\#$, define $\rho$ as the function that augments $(\mathcal{T}_{\mathcal{G}}, \mathcal{T}_{\mathcal{G}'})$ with blank trees:

$$\rho : (\mathcal{T}_{\mathcal{G}}^\#, \mathcal{T}_{\mathcal{G}'}^\#) \mapsto \left( \mathcal{T}_{\mathcal{G}}^\# \bigcup \mathcal{T}_{\mathbb{0}}^{\max(n,0)}, \mathcal{T}_{\mathcal{G}'}^\# \bigcup \mathcal{T}_{\mathbb{0}}^{\max(-n,0)} \right),$$

where $n = |\mathcal{T}_{\mathcal{G}'}^\#| - |\mathcal{T}_{\mathcal{G}}^\#|$.

Clearly, the leaf nodes of any node $\tilde{v}$ from a computation tree $T_v^L$ are its neighbor nodes $\mathcal{N}(\tilde{v})$ in the graph $\mathcal{G}$. Also, the number of elements in the multiset $\mathcal{T}_{\mathcal{G}}^{\#, L}$ equals to the number of nodes in $\mathcal{G}$. The role of $\rho$ is to ensure that the multisets $\mathcal{T}_{\mathcal{G}}^\#$ and $\mathcal{T}_{\mathcal{G}'}^\#$ have an equal number of elements, which is helpful since we need to compute the earth mover's distance between two multisets later.

Before moving on, let us review the definition of the earth mover's distance. Denote by $\mathcal{X}^\# = \{\{x_i\}\}_{i=1}^n$ and $\mathcal{Y}^\# = \{\{y_i\}\}_{i=1}^n$ two multisets containing $n$ elements, and $C \in \mathbb{R}^{n \times n}$ the cost matrix defined as $C_{ij} = d(x_i, y_j)$, where $d(\cdot, \cdot)$ is a predefined distance. The earth mover's distance is the solution of the following OT problem:

$$\begin{aligned} \text{OT}_d(\mathcal{X}, \mathcal{Y}) &= \min_{T \in \Gamma(X, Y)} \sum_{i,j} C_{ij} T_{ij}, \\ \Gamma(\mathcal{X}, \mathcal{Y}) &= \{\gamma \in \mathbb{R}_+^{n \times n} : \gamma \mathbb{1}_n = \gamma^\top \mathbb{1}_n = \mathbb{1}_n\}, \end{aligned} \tag{8}$$

where $T \in \mathbb{R}_+^{n \times n}$ is the transportation plan. $\mathbb{1}_n$ is a $n$-dimensional all-ones vector. Now we introduce another necessary notation that will be used in the proof later. Denote by $T$ a tree with root $r$. For any node $u \in T$ such that $u \neq r$, we use

$p(u)$ to denote its parent node. Let $\deg(u)$ be the degree of any non-leaf node $u \in T$, we define a function $d^*(\cdot)$ that map each node $u \in T$ into $\mathbb{N}$:

$$d^*(u) = \begin{cases} 1 & \text{if } u = r \\ \deg(p(u)) \cdot d^*(p(u)) & \text{otherwise} \end{cases}. \tag{9}$$

Clearly, $d^*(u)$ is the product of the degrees of the nodes along the shortest path from $u$ to the root $r$. Particularly, if $u = r$, we have $d(u) = 1$. With this notation, we introduce the the concept of modified tree distance, which is inspired by Definition 4 in (Chuang & Jegelka, 2022) and it measures the discrepancy between two computation trees.

**Definition D.3** (Modified Tree Distance). The tree distance between two trees $\mathcal{T}_a$ with root $r_a$ and $\mathcal{T}_b$ with root $r_b$ is

$$\text{TD}(\mathcal{T}_a, \mathcal{T}_b) = \begin{cases} \text{OT}_{\text{TD}}(\rho(\mathcal{T}_{r_a}^{\#}, \mathcal{T}_{r_b}^{\#})) & \text{if } L > 1 \\ \left\| \frac{\mathbf{x}_{r_a}}{d^*(r_a)} - \frac{\mathbf{x}_{r_b}}{d^*(r_b)} \right\| & \text{otherwise} \end{cases}, \tag{10}$$

where $L = \max(\text{Depth}(\mathcal{T}_a), \text{Depth}(\mathcal{T}_b))$. The features of $r_a$ and $r_b$ are denoted by $\mathbf{x}_{r_a}$ and $\mathbf{x}_{r_b}$, respectively. $\mathcal{T}_v^{\#}$ represents the multisets of computation trees whose root are the descendants of $v$.

Based on the tree distance, we present the modified definition of tree mover's distance, which is inherited from Definition 5 in (Chuang & Jegelka, 2022).

**Definition D.4** (Modified Tree Mover's Distance). For two graphs $\mathcal{G}_a$, $\mathcal{G}_b$ and $L \geq 1$, the tree mover's distance between $\mathcal{G}_a$ and $\mathcal{G}_b$ is defined as

$$\text{TMD}^L(\mathcal{G}_a, \mathcal{G}_b) = \text{OT}_{\text{TD}}(\rho(\mathcal{T}_{\mathcal{G}_a}^{\#,L}, \mathcal{T}_{\mathcal{G}_b}^{\#,L})), \tag{11}$$

where $\mathcal{T}_{\mathcal{G}_a}^{\#,L}$ and $\mathcal{T}_{\mathcal{G}_b}^{\#,L}$ are multisets of depth-$L$ computation trees of graphs $\mathcal{G}_a$ and $\mathcal{G}_b$, respectively.

With this definition, we analyze the Lipschitz constant of a spectral GNN with polynomial filter $h_\alpha$, as shown in the following theorem.

**Theorem D.5.** *Suppose that the activation function $\sigma(\cdot)$ is $K_\sigma$-Lipschitz and the graph readout function $\phi(\cdot)$ is $K_\phi$-Lipschitz. For a spectral GNN with output given by Eq. (6), we have*

$$\|h_\alpha(\mathcal{G}_a; w) - h_\alpha(\mathcal{G}_b; w)\| \leq \|\mathbf{W}_1\| \|\mathbf{W}_2\| K_\phi K_\sigma^2 \left( \sum_{k=0}^{K} |\alpha_k| \cdot \text{TMD}^{k+1}(\mathcal{G}_a, \mathcal{G}_b) \right), \tag{12}$$

*where $\mathbf{W}_1, \mathbf{W}_2$ are the weight matrices, and $\alpha_0, \ldots, \alpha_K$ are polynomial coefficients.*

Theorem D.5 shows that the difference in output of a spectral GNN on two graphs can be upper bounded by the summation of the tree mover's distance between them, ranging from order $0$ to $K$. Also, the upper bound includes the norm of weight matrices and polynomial coefficients, which reflects how these learnable parameters affect the stability of this spectral GNN. Notice that the analysis in (Chuang & Jegelka, 2022) does not consider these learnable parameters. Moreover, it is shown in (Chuang & Jegelka, 2022) that the tree mover's distance is a pseudometric. Following their proof process, it is easy to verify that the tree mover's distance we presented in Eq. (11) is also a pseudometric. Therefore, the summation $\sum_{k=0}^{K} \text{TMD}^{k+1}(\cdot, \cdot)$ is also a pseudometric. Suppose that $\|\mathbf{W}_1\|, \|\mathbf{W}_2\| \leq c_W$, we can rewrite Eq. (12) as

$$\|h_\alpha(\mathcal{G}_a; w) - h_\alpha(\mathcal{G}_b; w)\| \leq c_W^2 K_\phi K_\sigma^2 \left( \max_k |\alpha_k| \right) \left( \sum_{k=0}^{K} \text{TMD}^{k+1}(\mathcal{G}_a, \mathcal{G}_b) \right). \tag{13}$$

Therefore, the Lipschitz constant of a spectral GNN given by Eq. (6) is $(c_W^2 K_\phi K_\sigma^2 \max_k |\alpha_k|)/d_{\min}$ under the pseudometric $\sum_{k=0}^{K} \text{TMD}^{k+1}(\cdot, \cdot)$. This result indicates that a large absolute value of the coefficients $\max_k |\alpha_k|$ lead to a larger upper bound and thus could make the prediction of GNN being unstable. This finding is in accordance with that of Theorem 4.4, although the (pseudo)metric used to measure the discrepancy of input graphs are different.

## D.1. Proof of Theorem D.5

Recall that the output of the spectral GNN on graphs $\mathcal{G}_a, \mathcal{G}_b$ are

$$
\begin{aligned}
h_\alpha(\mathcal{G}_a, w) &= \phi\left(\sum_{i=1}^{n_a}[h_\alpha(\mathbf{D}_a^{-1}\mathbf{A}_a)\sigma(\sigma(\widetilde{\mathbf{X}}_a\mathbf{W}_1)\mathbf{W}_2)]_{i,:}\right), \\
h_\alpha(\mathcal{G}_b, w) &= \phi\left(\sum_{i=1}^{n_b}[h_\alpha(\mathbf{D}_b^{-1}\mathbf{A}_b)\sigma(\sigma(\widetilde{\mathbf{X}}_b\mathbf{W}_1)\mathbf{W}_2)]_{i,:}\right),
\end{aligned}
\tag{14}
$$

where $\hat{\mathbf{L}}_a$ and $\hat{\mathbf{L}}_b$ are the graph Laplacian matrix of $\mathcal{G}_a$ and $\mathcal{G}_b$. Also, $\widetilde{\mathbf{X}}_a$ and $\widetilde{\mathbf{X}}_b$ are the input features of $\mathcal{G}_a$ and $\mathcal{G}_b$. Let us define $\mathbf{Z}_a^{(0)} = \sigma(\sigma(\widetilde{\mathbf{X}}_a\mathbf{W}_1)\mathbf{W}_2)$ and $\mathbf{Z}_b^{(0)} = \sigma(\sigma(\widetilde{\mathbf{X}}_b\mathbf{W}_1)\mathbf{W}_2)$, then Eq. (14) can be rewritten as

$$
\begin{aligned}
h_\alpha(\mathcal{G}_a, w) &= \phi\left(\sum_{i=1}^{n_a}h_\alpha(\mathbf{D}_a^{-1}\mathbf{A}_a)\mathbf{Z}_{a,i,:}^{(0)}\right) = \phi\left(\sum_{i=1}^{n}\sum_{k=0}^{K}\alpha_k(\mathbf{D}_a^{-1}\mathbf{A}_a)^k\mathbf{Z}_{a,i,:}^{(0)}\right), \\
h_\alpha(\mathcal{G}_b, w) &= \phi\left(\sum_{i=1}^{n_b}h_\alpha(\mathbf{D}_b^{-1}\mathbf{A}_b)\mathbf{Z}_{b,i,:}^{(0)}\right) = \phi\left(\sum_{i=1}^{n}\sum_{k=0}^{K}\alpha_k(\mathbf{D}_b^{-1}\mathbf{A}_b)^k\mathbf{Z}_{b,i,:}^{(0)}\right).
\end{aligned}
\tag{15}
$$

Therefore, the difference in outputs can be upper-bounded by

$$
\begin{aligned}
\|h_\alpha(\mathcal{G}_a, w) - h_\alpha(\mathcal{G}_b, w)\| &= \left\|\phi\left(\sum_{i=1}^{n_a}\sum_{k=0}^{K}\alpha_k(\mathbf{D}_a^{-1}\mathbf{A}_a)^k\mathbf{Z}_{a,i,:}^{(0)}\right) - \phi\left(\sum_{i=1}^{n_b}\sum_{k=0}^{K}\alpha_k(\mathbf{D}_b^{-1}\mathbf{A}_b)^k\mathbf{Z}_{b,i,:}^{(0)}\right)\right\| \\
&\leq K_\phi\left\|\sum_{i=1}^{n_a}\sum_{k=0}^{K}\alpha_k(\mathbf{D}_a^{-1}\mathbf{A}_a)^k\mathbf{Z}_{a,i,:}^{(0)} - \sum_{i=1}^{n_b}\sum_{k=0}^{K}\alpha_k(\mathbf{D}_b^{-1}\mathbf{A}_b)^k\mathbf{Z}_{b,i,:}^{(0)}\right\| \\
&= K_\phi\left\|\sum_{k=0}^{K}\alpha_k\left(\sum_{i=1}^{n_a}(\mathbf{D}_a^{-1}\mathbf{A}_a)^k\mathbf{Z}_{a,i,:}^{(0)} - \sum_{i=1}^{n_b}(\mathbf{D}_b^{-1}\mathbf{A}_b)^k\mathbf{Z}_{b,i,:}^{(0)}\right)\right\| \\
&\leq K_\phi\sum_{k=0}^{K}|\alpha_k|\left\|\sum_{i=1}^{n_a}(\mathbf{D}_a^{-1}\mathbf{A}_a)^k\mathbf{Z}_{a,i,:}^{(0)} - \sum_{i=1}^{n_b}(\mathbf{D}_b^{-1}\mathbf{A}_b)^k\mathbf{Z}_{b,i,:}^{(0)}\right\|.
\end{aligned}
\tag{16}
$$

For any $k \in [K]$, define

$$
\Delta_k = \left\|\sum_{i=1}^{n_a}(\mathbf{D}_a^{-1}\mathbf{A}_a)^k\mathbf{Z}_{a,i,:}^{(0)} - \sum_{i=1}^{n_b}(\mathbf{D}_b^{-1}\mathbf{A}_b)^k\mathbf{Z}_{b,i,:}^{(0)}\right\|,
\tag{17}
$$

now it is sufficient to provide an upper bound for $\Delta_k$. Let $\mathcal{V}_a$ and $\mathcal{V}_b$ be the node sets of $\mathcal{G}_a$ and $\mathcal{G}_b$. Denote by $\mathcal{V}_a^\rho$ and $\mathcal{V}_b^\rho$ two multisets of nodes after blank tree augmentation, that is, $(\mathcal{V}_a^\rho, \mathcal{V}_b^\rho) = \rho(\mathcal{V}_a, \mathcal{V}_b)$. Then, the embedding sets $\{(\mathbf{D}_a^{-1}\mathbf{A}_a)^k\mathbf{Z}_{a,i,:}^{(0)}\}_{i=1}^{n_a}$ can be represented as $\{\mathbf{z}_i^{(k)}\}_{i\in\mathcal{V}_a}$. Similarly, we have $\{(\mathbf{D}_b^{-1}\mathbf{A}_b)^k\mathbf{Z}_{b,i,:}^{(0)}\}_{i=1}^{n_b} = \{\mathbf{z}_i^{(k)}\}_{i\in\mathcal{V}_b}$. Let $T^{(K,\mathcal{V}_a^\rho\mathcal{V}_b^\rho)}$ be the transport plan that aligns two multisets $\mathcal{V}_a^\rho$ and $\mathcal{V}_b^\rho$, then we have

$$
\begin{aligned}
\Delta_K &= \left\|\sum_{i=1}^{n_a}(\mathbf{D}_a^{-1}\mathbf{A}_a)^K\mathbf{Z}_{a,i,:}^{(0)} - \sum_{i=1}^{n_b}(\mathbf{D}_b^{-1}\mathbf{A}_b)^K\mathbf{Z}_{b,i,:}^{(0)}\right\| = \left\|\sum_{i\in\mathcal{V}_a}\mathbf{z}_i^{(K)} - \sum_{j\in\mathcal{V}_b}\mathbf{z}_j^{(K)}\right\| \\
&= \left\|\sum_{i\in\mathcal{V}_a^\rho, j\in\mathcal{V}_b^\rho}T_{i,j}^{(K,\mathcal{V}_a^\rho\mathcal{V}_b^\rho)}\left(\mathbf{z}_i^{(K)} - \mathbf{z}_j^{(K)}\right)\right\| \\
&\leq \sum_{i\in\mathcal{V}_a^\rho, j\in\mathcal{V}_b^\rho}T_{i,j}^{(K,\mathcal{V}_a^\rho\mathcal{V}_b^\rho)}\left\|\mathbf{z}_i^{(K)} - \mathbf{z}_j^{(K)}\right\|.
\end{aligned}
\tag{18}
$$

Next, we establish a concrete formulation for the embedding $\mathbf{z}^{(K)}$. Notice that the embedding $\mathbf{z}^{(K)}$ is obtained from the following message passing rules:

$$
\mathbf{z}_v^{(l)} = \sum_{u\in\mathcal{N}(v)}\frac{\mathbf{z}_u^{(l-1)}}{d_v}, \ell = 1, \ldots, K, \mathbf{z}_v^{(0)} = \sigma(\sigma(\widetilde{\mathbf{X}}_v\mathbf{W}_1)\mathbf{W}_2).
\tag{19}
$$

Then we have

$$\sum_{i_1 \in \mathcal{V}_a^\rho, j_1 \in \mathcal{V}_b^\rho} T_{i_1,j_1}^{(K,\mathcal{V}_a^\rho \mathcal{V}_b^\rho)} \left\| \mathbf{z}_{i_1}^{(K)} - \mathbf{z}_{j_1}^{(K)} \right\|$$

$$= \sum_{i_1 \in \mathcal{V}_a^\rho, j_1 \in \mathcal{V}_b^\rho} T_{i_1,j_1}^{(K,\mathcal{V}_a^\rho \mathcal{V}_b^\rho)} \left\| \sum_{i_2 \in \mathcal{N}(i_1)} \frac{\mathbf{z}_{i_2}^{(K-1)}}{d_{i_1}} - \sum_{j_2 \in \mathcal{N}(j_1)} \frac{\mathbf{z}_{j_2}^{(K-1)}}{d_{j_1}} \right\|$$

$$= \sum_{i_1 \in \mathcal{V}_a^\rho, j_1 \in \mathcal{V}_b^\rho b} T_{i_1,j_1}^{(K,\mathcal{V}_a^\rho \mathcal{V}_b^\rho)} \left\| \sum_{i_2 \in \mathcal{N}^\rho(i_1), j_2 \in \mathcal{N}^\rho(j_1)} T_{i_2,j_2}^{(K-1,\mathcal{N}^\rho(i_1)\mathcal{N}^\rho(j_1))} \left( \frac{\mathbf{z}_{i_2}^{(K-1)}}{d_{i_1}} - \frac{\mathbf{z}_{j_2}^{(K-1)}}{d_{j_1}} \right) \right\|$$

$$\leq \sum_{i_1 \in \mathcal{V}_a^\rho, j_1 \in \mathcal{V}_b^\rho b} T_{i_1,j_1}^{(K,\mathcal{V}_a^\rho \mathcal{V}_b^\rho)} \sum_{i_2 \in \mathcal{N}^\rho(i_1), j_2 \in \mathcal{N}^\rho(j_1)} T_{i_2,j_2}^{(K-1,\mathcal{N}^\rho(i_1)\mathcal{N}^\rho(j_1))} \left\| \frac{\mathbf{z}_{i_2}^{(K-1)}}{d_{i_1}} - \frac{\mathbf{z}_{j_2}^{(K-1)}}{d_{j_1}} \right\|$$

$$= \sum_{i_1 \in \mathcal{V}_a^\rho, j_1 \in \mathcal{V}_b^\rho b} T_{i_1,j_1}^{(K,\mathcal{V}_a^\rho \mathcal{V}_b^\rho)} \sum_{i_2 \in \mathcal{N}^\rho(i_1), j_2 \in \mathcal{N}^\rho(j_1)} T_{i_2,j_2}^{(K-1,\mathcal{N}^\rho(i_1)\mathcal{N}^\rho(j_1))} \left\| \sum_{i_3 \in \mathcal{N}(i_2)} \frac{\mathbf{z}_{i_3}^{(K-2)}}{d_{i_1} d_{i_2}} - \sum_{j_3 \in \mathcal{N}(j_2)} \frac{\mathbf{z}_{j_2}^{(K-2)}}{d_{j_1} d_{j_2}} \right\|$$

$$= \sum_{i_1 \in \mathcal{V}_a^\rho, j_1 \in \mathcal{V}_b^\rho b} T_{i_1,j_1}^{(K,\mathcal{V}_a^\rho \mathcal{V}_b^\rho)} \sum_{i_2 \in \mathcal{N}^\rho(i_1), j_2 \in \mathcal{N}^\rho(j_1)} T_{i_2,j_2}^{(K-1,\mathcal{N}^\rho(i_1)\mathcal{N}^\rho(j_1))}$$

$$\cdot \left\| \sum_{i_3 \in \mathcal{N}^\rho(i_2), j_3 \in \mathcal{N}^\rho(j_2)} T_{i_3,j_3}^{(K-2,\mathcal{N}^\rho(i_2)\mathcal{N}^\rho(j_2))} \left( \frac{\mathbf{z}_{i_3}^{(K-2)}}{d_{i_1} d_{i_2}} - \frac{\mathbf{z}_{j_3}^{(K-2)}}{d_{j_1} d_{j_2}} \right) \right\|$$

$$\leq \sum_{i_1 \in \mathcal{V}_a^\rho, j_1 \in \mathcal{V}_b^\rho b} T_{i_1,j_1}^{(K,\mathcal{V}_a^\rho \mathcal{V}_b^\rho)} \sum_{i_2 \in \mathcal{N}^\rho(i_1), j_2 \in \mathcal{N}^\rho(j_1)} T_{i_2,j_2}^{(K-1,\mathcal{N}^\rho(i_1)\mathcal{N}^\rho(j_1))}$$

$$\cdot \sum_{i_3 \in \mathcal{N}^\rho(i_2), j_3 \in \mathcal{N}^\rho(j_2)} T_{i_3,j_3}^{(K-2,\mathcal{N}^\rho(i_2)\mathcal{N}^\rho(j_2))} \left\| \frac{\mathbf{z}_{i_3}^{(K-2)}}{d_{i_1} d_{i_2}} - \frac{\mathbf{z}_{j_3}^{(K-2)}}{d_{j_1} d_{j_2}} \right\|.$$

By recursively expand the rest terms, we have

$$\sum_{i \in \mathcal{V}_a^\rho, j \in \mathcal{V}_b^\rho} T_{i,j}^{(K,\mathcal{V}_a^\rho \mathcal{V}_b^\rho)} \left\| \mathbf{z}_i^{(K)} - \mathbf{z}_j^{(K)} \right\|$$

$$\leq \sum_{i_1 \in \mathcal{V}_a^\rho, j_1 \in \mathcal{V}_b^\rho} T_{i,j}^{(K,\mathcal{V}_a^\rho \mathcal{V}_b^\rho)} \cdots \sum_{i_{K+1} \in \mathcal{N}^\rho(i_K), j_{K+1} \in \mathcal{N}^\rho(j_K)} T_{i_{K+1},j_{K+1}}^{(0,\mathcal{N}^\rho(i_K)\mathcal{N}^\rho(j_K))} \left\| \frac{\mathbf{z}_{i_{K+1}}^{(0)}}{\prod_{k=1}^K d_{i_k}} - \frac{\mathbf{z}_{j_{k+1}}^{(0)}}{\prod_{k=1}^K d_{j_k}} \right\|$$

$$\leq \|\mathbf{W}_1\| \|\mathbf{W}_2\| K_\sigma^2 \sum_{i_1 \in \mathcal{V}_a^\rho, j_1 \in \mathcal{V}_b^\rho} T_{i,j}^{(K,\mathcal{V}_a^\rho \mathcal{V}_b^\rho)} \cdots \sum_{i_{K+1} \in \mathcal{N}^\rho(i_K), j_{K+1} \in \mathcal{N}^\rho(j_K)} T_{i_{K+1},j_{K+1}}^{(0,\mathcal{N}^\rho(i_K)\mathcal{N}^\rho(j_K))} \left\| \frac{\widetilde{\mathbf{X}}_{i_{K+1}}}{\prod_{k=1}^K d_{i_k}} - \frac{\widetilde{\mathbf{X}}_{j_{k+1}}}{\prod_{k=1}^K d_{j_k}} \right\|$$

$$= \|\mathbf{W}_1\| \|\mathbf{W}_2\| K_\sigma^2 \text{TMD}^{K+1}(\mathcal{G}_a, \mathcal{G}_b). \tag{20}$$

By the same way, we have

$$\Delta_k \leq \|\mathbf{W}_1\| \|\mathbf{W}_2\| K_\sigma^2 \text{TMD}^{k+1}(\mathcal{G}_a, \mathcal{G}_b), k = 1, \ldots, K. \tag{21}$$

Combining Eq. (16) and Eq. (21) we have

$$\|h_\alpha(\mathcal{G}_a, w) - h_\alpha(\mathcal{G}_b, w)\| \leq \|\mathbf{W}_1\| \|\mathbf{W}_2\| K_\phi K_\sigma^2 \sum_{k=0}^K |\alpha_k| \text{TMD}^{K+1}(\mathcal{G}_a, \mathcal{G}_b). \tag{22}$$

This finishes the proof. It is worth mentioning that we can derive similar upper bound for the difference of node-level output $\|h_\alpha(v_i, w) - h_\alpha(v_j, w)\|$, where $v_i, i \in [n_a]$ and $v_j, j \in [n_b]$ are nodes from $\mathcal{G}_a$ and $\mathcal{G}_b$, respectively. The definitions of $h_\alpha(v_i, w)$ and $h_\alpha(v_j, w)$ are as follows

$$h_\alpha(v_i, w) = [h_\alpha(\mathbf{D}_a^{-1} \mathbf{A}_a) \sigma(\sigma(\widetilde{\mathbf{X}}_a \mathbf{W}_1) \mathbf{W}_2)]_{i,:},$$
$$h_\alpha(v_j, w) = [h_\alpha(\mathbf{D}_b^{-1} \mathbf{A}_b) \sigma(\sigma(\widetilde{\mathbf{X}}_b \mathbf{W}_1) \mathbf{W}_2)]_{j,:}. \tag{23}$$

By the same way, we have

$$\|h_\alpha(v_i, w) - h_\alpha(v_j, w)\| \leq \|\mathbf{W}_1\|\|\mathbf{W}_2 K_\sigma^2 \sum_{k=0}^{K} |\alpha_k| \mathrm{OT}_{\mathrm{TD}}(\rho(\mathcal{T}_{v_i}^{\#,k+1}, \mathcal{T}_{v_j}^{\#,k+1})). \tag{24}$$

# E. Additional Eigenvector Visualizations

## E.1. Horse Mesh Eigenvectors

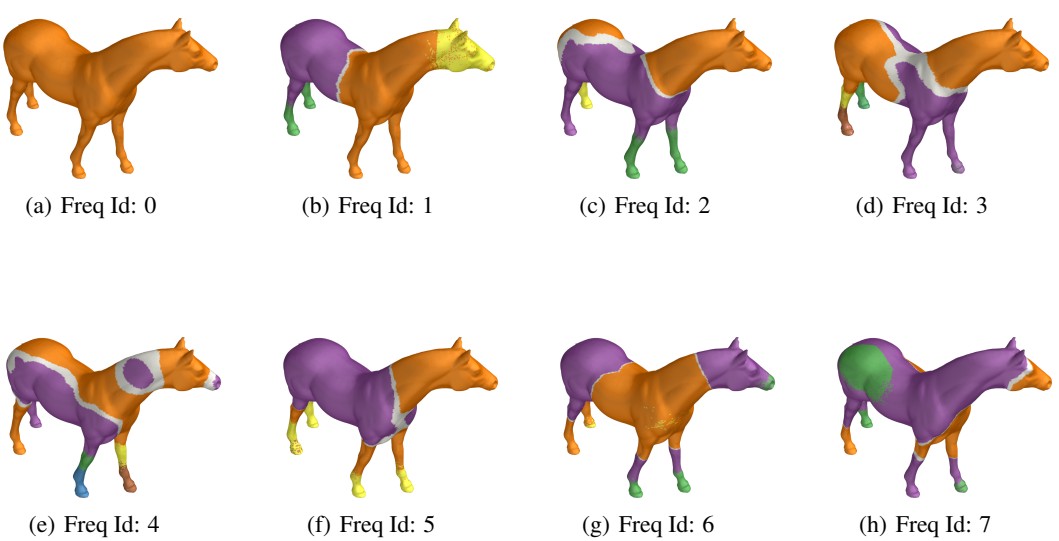

(a) Freq Id: 0    (b) Freq Id: 1    (c) Freq Id: 2    (d) Freq Id: 3

(e) Freq Id: 4    (f) Freq Id: 5    (g) Freq Id: 6    (h) Freq Id: 7

*Figure 5.* Low-frequency eigenvectors on horse mesh (first 8). Note the smooth, global patterns that vary gradually across the mesh.

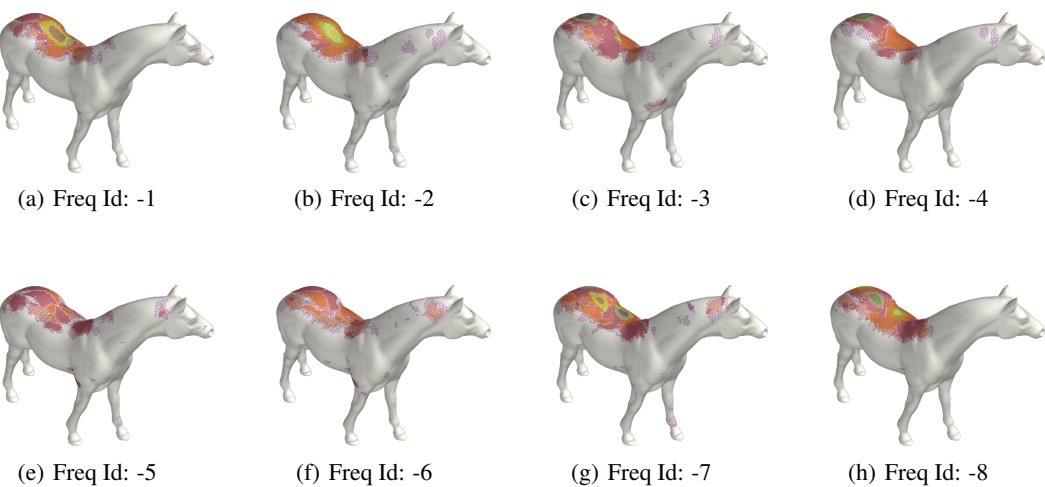

(a) Freq Id: -1    (b) Freq Id: -2    (c) Freq Id: -3    (d) Freq Id: -4

(e) Freq Id: -5    (f) Freq Id: -6    (g) Freq Id: -7    (h) Freq Id: -8

*Figure 6.* High-frequency eigenvectors on horse mesh (last 8). Note the localized behavior and concentration of energy in small regions.

## E.2. Sphere Mesh Eigenvectors

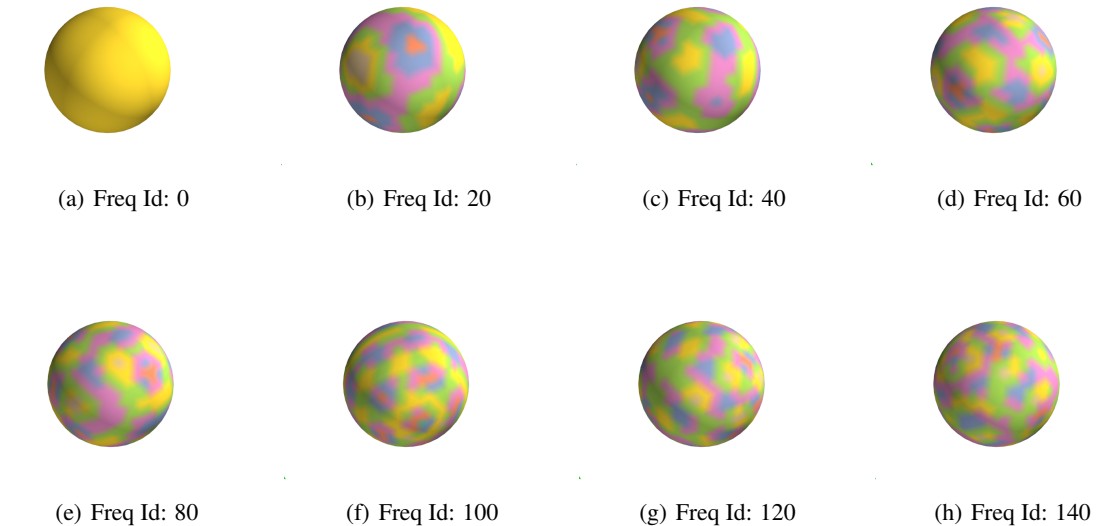

(a) Freq Id: 0      (b) Freq Id: 20      (c) Freq Id: 40      (d) Freq Id: 60

(e) Freq Id: 80      (f) Freq Id: 100      (g) Freq Id: 120      (h) Freq Id: 140

*Figure 7.* Low-frequency eigenvectors on sphere mesh (selected). Note the smooth, global patterns similar to spherical harmonics.

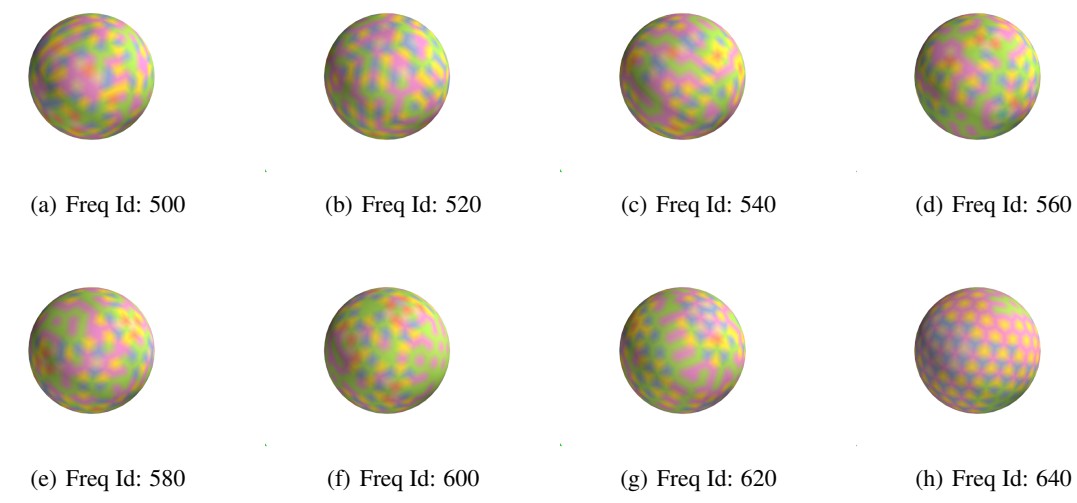

(a) Freq Id: 500      (b) Freq Id: 520      (c) Freq Id: 540      (d) Freq Id: 560

(e) Freq Id: 580      (f) Freq Id: 600      (g) Freq Id: 620      (h) Freq Id: 640

*Figure 8.* High-frequency eigenvectors on sphere mesh (selected). Note the highly localized patterns.

# F. Theorems

## F.1. Theorems in Section 3.4

The following theorems provide supports that low-frequency Laplacian eigenvectors and Laplacian eigenvectors on special graphs indeed possess "semantic interpretability".

**Cheeger's inequality** leads to the conclusion that $\lambda_2$ reflects the graph's connectivity, and the eigenvector $\mathbf{u}_2$ can reflect the global community structure of the graph.

**Theorem F.1** (Cheeger's Inequality (Chung, 1996)). *$\lambda_2$ is related to the Cheeger's constant of $h(\mathcal{G})$ by: $\frac{\lambda_2}{2} \leq h(\mathcal{G}) \leq \sqrt{2\lambda_2}$, where $h(\mathcal{G}) = \min_{\mathcal{S} \subset \mathcal{V}} \frac{|E(\mathcal{S}, \bar{\mathcal{S}})|}{d|\mathcal{S}|}$, and $E(\mathcal{S}, \bar{\mathcal{S}})$ denotes the set of edges between $\mathcal{S}$ and its complement, $d$ is the degree, $|\mathcal{S}|$*

*is the size of $\mathcal{S}$.*

*Remark* F.2. (1).The selected partition $(\mathcal{S}/\hat{\mathcal{S}})$ relies on $\mathbf{u}_2$ by *sorting* $\{\mathbf{u}_{2,i}\}_{i=1}^N$ . (2).The theorem extends to other low-frequency eigenvectors (Lee et al., 2014).

Another strong theoretical result is the **Nodal Domain Theorem**.

**Theorem F.3** (Nodal Domain Theorem (Davies et al., 2000)). *The number of nodal domains of $\mathbf{u}_k$, i.e., the maximal connected subgraphs where $\mathbf{u}_k$ does not change sign, satisfies: the number of nodal domains of $\mathbf{u}_k$ is at most $k$. Furthermore, there exists at least one eigenvector corresponding to $\lambda_k$ that has exactly $k$ nodal domains.*

The following theorem gives the eigenvectors of path and cycle graphs.

**Theorem F.4** (Path and Cycle Graph Eigenvectors). *For a path graph with $n$ vertices, one possible choice of orthonormal eigenvectors is: $u_k(n) = \sqrt{\frac{2}{n}} \cos(\frac{\pi k(n-0.5)}{n})$, $k = 1, ..., n-1$. For a cycle graph with $n$ vertices, the eigenvectors form the DFT matrix columns: $u_k = \frac{1}{\sqrt{n}}[1, \omega^k, \omega^{2k}, ..., \omega^{(n-1)k}]^T$, $\omega = e^{2\pi i/n}$.*

The following theorem gives the eigenvectors of the Cartesian product of graphs.

**Theorem F.5** (Eigenvectors of Cartesian Product of Graphs). *Let $\mathcal{G}_1$ and $\mathcal{G}_2$ be two graphs with adjacency matrices $\mathbf{A}_1$ and $\mathbf{A}_2$, respectively. Denote their Cartesian product's adjacency matrix as $\mathbf{A}_\square$. If $\lambda_1$ and $\lambda_2$ are eigenvalues of $\mathbf{A}_1$ and $\mathbf{A}_2$ with eigenvectors $\mathbf{x}_1$ and $\mathbf{x}_2$, then $\lambda_1 + \lambda_2$ is an eigenvalue of $\mathbf{A}_\square$ with eigenvector $\mathbf{x}_1 \otimes \mathbf{x}_2$.*

### F.2. Theorem in Section 4.5

The following result gives the upper bound for the transductive generalization gap.

**Theorem F.6** (Tang & Liu, 2023, Proposition 4.15). *Suppose that $\mathbf{x}_i \leq c_X$ holds for any $i \in [n]$. For a spectral GNN with output given by Eq. (7), let $w_0$ be its initial parameter and $\mathcal{W}_R = \{w : \|w - w_0\| \leq R\}$ be the parameter space. Suppose that $\|\mathbf{W}_1\|, \|\mathbf{W}_2\| \leq c_W$ hols for any $w \in \mathcal{W}$, and the activation function $\sigma(\cdot)$ is $K_\sigma$-Lipschitz. For any $\delta \in (0,1)$, with probability as least $1 - \delta$ over the randomness of $\pi$, we have*

$$R_{\text{test}}(w, \pi) - R_{\text{train}}(w, \pi) = \mathcal{O}\left(\frac{(m+u)^{\frac{3}{2}}R\sqrt{L_1^2 + L_2^2}}{mu}\right),$$

*where*

$$L_1 = \sqrt{2}c_X c_W^2 K_\sigma^2 \left(\sum_{k=0}^{K} \|\hat{\mathbf{L}}^k\|_\infty\right)^{\frac{1}{2}}, \quad L_2 = 2c_X c_W K_\sigma^2 \|h_\alpha(\hat{\mathbf{L}})\|_\infty. \tag{25}$$

