# OpenReview forum: "Position: Spectral GNNs Rely Less on Graph Fourier Basis than Conceived"
_ICML.cc/2025/Position_Paper_Track — ICML 2025 Position Paper Track poster_

### Official Review · Reviewer_s9LH · 2025-02-26

**Significance:** 2
**Argument Clarity:** 2
**Rating:** 3
**Confidence:** 3

**Questions:**

---

**Discussion Potential:**

2

**Paper Summary:**

The paper deals with spectral graph learning. It questions the common belief that spectral GNNs are based on the graph Fourier basis. Moreover, the paper discusses the limitations of expressive filters for stability and generalization.  The authors challenge the conventional understanding that spectral GNNs derive their effectiveness from the Graph Fourier Transform (GFT). They identify two key issues: (1) the questionable role of graph Laplacian eigenvectors as a Fourier basis, as they do not necessarily preserve the semantic properties of classical Fourier analysis, and (2) the fundamental limitations of polynomial filters, which naturally limit expressiveness to ensure stability and generalization. The paper argues that these issues are intertwined, as the stability constraints obscure the shortcomings of the Fourier basis assumption. The authors provide theoretical analysis and empirical evidence demonstrating the localization of high-frequency eigenvectors, which contradicts their assumed global oscillatory nature. They also show that highly expressive spectral filters lead to instability and poor generalization, necessitating simpler polynomial approximations. This challenges the notion that spectral GNNs fundamentally operate in the spectral domain, suggesting a need for alternative graph signal dictionaries and a re-examination of polynomial filtering techniques. The work ultimately calls for a reassessment of spectral GNNs' theoretical foundations to better align with their practical effectiveness.

## update after rebuttal

I slightly increased my score.

**Position:**

Yes

**Position In Title:**

Yes

**Related Work:**

2

**Strengths And Weaknesses:**

**Strengths**
- Challenges a widely accepted assumption in spectral GNNs, providing a new perspective on the role of the GFT and possibly leading to deeper theoretical investigations.
- I liked the discussion on why highly expressive spectral filters lead to instability and poor generalization
- The authors support their claims with a theoretical analysis

**Weaknesses**
- The paper should put more work into outlining the background behind spectral GNNs. The presentation in Table 2 is slightly obscure. Why not include a proper derivation of spectral GNNs?
- The alternative view section is very generic and does not provide much insights
- While the paper critiques existing spectral GNN foundations, it does not provide concrete replacements for the Graph Fourier basis, leaving the reader without a clear next step.
- The paper has several typos, and some sentences are hard to parse.

**Support:**

3

---

> ### Author Rebuttal · Authors · 2025-04-01
>
> Dear Reviewer:
>
> Thank you for your suggestions! We will address each point mentioned in the **Weakness** section.
>
> > **W1:** The paper should put more work into **outlining the background behind spectral GNNs**. The presentation in Table 2 is slightly obscure. Why not include a proper derivation of spectral GNNs?
>
> Thank you for your suggestion.  While we previously aimed to be comprehensive while maintaining conciseness, "Section 2.1 Background - Polynomial filters" does indeed progress rather quickly. We plan to add a detailed description in the appendix.
>
> > **W2:** The **alternative view** section is very generic and does not provide much insights
>
> Thank you for sharing this impression.
> The reason we wrote this section briefly is that the **entire paper** revolves around deconstructing this alternative view.
>
>
>
> When we were writing, we realized that our alternative view was often taken as "self-evident" - it's not a strikingly novel perspective, but rather something that has been unconsciously internalized in the community. This is why **we spent considerable space in the earlier sections,  analyzing how this "alternative view" came to be taken for granted**, and consequently kept this final section very brief.
>
> Our alternative view is a seemingly "natural" belief: **first**,  graph Fourier bases are a natural extension of traditional Fourier bases; **second**,  we use polynomials to achieve computability； **together**, these lead to polynomial filter models, which are a "compromised version" of graph convolution that doesn't fully utilize its expressive power.
>
> Why do we say that we've been analyzing the alternative view throughout the earlier sections?
>
> 1. In the **Background** chapter, particularly **Table 1**, we present the general "mathematical foundation" that the alternative view is based on in **GSP** - something that we found was not mentioned in almost all Spectral GNN works we reviewed, but which we have clearly listed. In **Section 3.1**, through this correspondence, we clearly see that "while the translation of mathematical concepts appears rigorous, it is also rough (such as the correspondence of gradients), so although graph Fourier bases and Fourier bases are both eigenfunctions of the Laplacian, their semantics may not be directly translatable."
>
> 2. Also in the Background chapter, we wrote about how the alternative view introduced polynomials. You criticized that we didn't write this part specifically enough, and we will make modifications. Thank you again for this point.
>
> 3. In **Section 3.4**, we also deeply explored how **special cases**, **generalization tendencies** in thinking, and related research history influenced the formation of the alternative view.
>
> In summary, when writing, **we believed that our strength lay in carefully deconstructing the alternative view**, which is why we kept the section required by **CFP** very simple.
>
> Thank you for pointing this out. We will improve this aspect.
>
> > **W3:** While the paper critiques existing spectral GNN foundations, it does not provide concrete replacements for the **Graph Fourier basis**, leaving the reader without a clear next step.
>
> We believe that providing "concrete alternatives" is not a mandatory requirement for a **position paper**.
>
> > **W4:** The paper has several **typos**, and some sentences are hard to parse.
>
> Thank you for pointing this out. We will check again.
>
>
> Best regards,
>
> the authors

---

> > ### Comment · Reviewer_s9LH · 2025-04-08
> >
> > The rebuttal does not address my points of criticism but tries to circumvent them, so I will keep my score for now.

---

> > > ### Author Response · Authors · 2025-04-09
> > >
> > > Dear Reviewer.
> > >
> > > Sorry for giving the impression that we were avoiding the issue. And thank you for giving us another chance to respond.
> > >
> > > **I. Writing of alternative views.**
> > >
> > > Your main criticism seems to be that: **alternative view is not sufficiently developed in the writing**. We tried to argue that the way we structured our argument was actually intended to *progressively build up* this alternative view throughout the earlier sections. Without such buildup, it would have been difficult for us to clearly articulate the issues we wanted to highlight—especially because these issues are quite subtle. As one of the previous reviewers noted, we have indeed maintained a “focused argument throughout.” That said, we appreciate your feedback—**our writing should certainly better meet expectations and present a more complete structure. We hope to have the opportunity to revise further**.
> > >
> > > **II. Discussion of alternative methods.**
> > >
> > > Another important criticism is that we didn’t propose **concrete alternative methods**. We still believe this goes somewhat beyond our current ability and the intended scope of a position paper. If try to provide some preliminary ideas, we would include them in the discussion section:
> > >
> > > -  **Investigate why current polynomial-based GNNs seem to work well**  from other aspects other than GSP. Could it be related to the convergence properties of linear GNNs (e.g., as discussed in Xu et al., 2021)? Is the use of polynomial bases related to numerical stability?
> > >
> > > - If we still adhere to the graph signal perspective, we should first develop more precise **probing methods** for whether the GFB we are using is good, and whether they are semantically well sounded. Then, we could **explore a broader range of graph dictionaries**—though they may suffer from **computational complexity** (e.g., the windowed GFB proposed by Shuman et al., 2013). Perhaps combining these with hierarchical coarsening techniques could help mitigate the computational cost.
> > >
> > > - For certain types of graphs where the graph Fourier basis carries clear semantic meaning, we could explore decomposition approaches—such as treating the graph as a composition of circulant graphs, like in this paper: https://www2.eecs.berkeley.edu/Pubs/TechRpts/2013/EECS-2013-209.pdf.
> > >
> > > Do you think such discussion that is quite general can serve as a start point for finding alternative?
> > >
> > > Thanks!
> > >
> > > the Authors

---

### Official Review · Reviewer_eTxu · 2025-03-13

**Significance:** 3
**Argument Clarity:** 2
**Rating:** 3
**Confidence:** 5

**Questions:**

Do you think the field of Spectral GNNs has misinterpreted or oversimplified these prior results?

What are some specific misconceptions that need to be corrected in future research?

Should we abandon spectral filtering approaches altogether, or is there a way to redefine them in a more theoretically sound way? Could alternative graph dictionaries (as you mention) provide a better foundation? What would be a good starting point for exploring them?

**Discussion Potential:**

4

**Paper Summary:**

The paper critically examines two fundamental assumptions about Spectral GNNs:

1. **The validity of the Graph Fourier basis**
   - It questions whether graph Laplacian eigenvectors truly serve as a Fourier basis, arguing that the transition from continuous Fourier analysis to graph-based Fourier analysis does not fully preserve semantic properties.
   - The authors present theoretical and empirical evidence showing that high-frequency eigenvectors behave differently from their classical Fourier counterparts, challenging the assumption that Spectral GNNs operate in a frequency-specific manner.

2. **The limitations of expressive spectral filters**
   - The paper argues that while more expressive spectral filters might seem beneficial, they actually reduce stability and generalization, making them impractical for real-world applications.
   - It demonstrates that the avoidance of complex filters has indirectly prevented researchers from fully questioning the role of the Graph Fourier basis.


### update after rebuttal
Q1 response: The mentioned paper lacks a comprehensive theoretical examination of spectral graph neural networks. While the experimental results may suggest certain implications, they necessitate a more robust theoretical analysis for validation.

I raised my score because of the overall rebuttal.

**Position:**

Yes

**Position In Title:**

Yes

**Related Work:**

2

**Strengths And Weaknesses:**

**Strengths**

Provides empirical evidence
- The authors present visualizations and numerical experiments that reveal discrepancies between classical Fourier analysis and the behavior of graph eigenvectors.
- Their experiments highlight the localization of high-frequency eigenvectors, showing that they do not exhibit the expected global oscillatory patterns.



**Weaknesses**
No concrete alternative is proposed
- The paper effectively questions existing assumptions but does not offer a clear alternative framework for spectral graph learning.
- It suggests exploring other graph dictionaries but does not provide a concrete method or model to replace the Graph Fourier basis.

Limited impact on practical applications
- While the theoretical discussion is insightful, the paper does not demonstrate the practical consequences of its findings.
- It does not analyze whether removing the Graph Fourier basis from Spectral GNNs leads to better architectures.

**Support:**

3

---

> ### Author Rebuttal · Authors · 2025-04-01
>
> Dear Reviewer:
>
> Thank you for your valuable feedback on our work. We would first discuss the questions you raised, as we believe these discussions could significantly enrich our paper. Then we will respond to the weakness part---we maintain that concrete alternative solutions and practical application impacts are not requirements for the Position Paper Track.
>
> ---
>
> **I. Discussion with Questions**
>
> > **Q1:** Do you think the field of Spectral GNNs has misinterpreted or oversimplified these prior results?
>
> Yes. This stems from an **uncritical belief** in the properties of medium and high-frequency Fourier bases, combined with the inherent difficulty in directly observing eigenbasis patterns in graph visualization.
>
> Another reason of  misinterpretion might stem from the different hyperparameter tuning techniques (from the crude tuning in ChebNet to the widespread use of tools like Optuna in modern architectures like JacobiConv and OptBasisGNN) has also made some earlier conclusions less reliable. For example, a recent [paper](https://openreview.net/pdf?id=xkljKdGe4E) demonstrated that with careful tuning, normalization, and residual connections, GCN can outperform spectral GNNs on heterophilic graphs, directly challenging the widely-held belief that "GCN is a low-frequency filter, while heterophilic graphs need high-frequency Fourier bases".
>
> > **Q2:** What are some **specific misconceptions** that need to be corrected?
>
> **Misconception 1:** Laplacian eigenbasis generally poses Fourier basis properties, and should be manipulated (by polynomial).
>
> **Recommended approach:** (1) The researchers can limit confidence to **specific** scenarios like low-frequency components and well-structured graphs; (2) When using GFB, one can **evaluate the semantic reasonableness** of bases $\\{u_i\\}_i$. The
> Localization and Compressibility mentioned in
> Section 3.2 can serve as a starting point.
> For reference, Dorina Thanou et al.'s work
> "Learning Parametric Dictionaries for Signals
> on Graphs" explicitly considers Compressibility.
>
> **Misconception 2:** The role of polynomial  bases are to compromisively approximation the "the ideal filter", and this is why polynomial filters work.
>
> **Recommended approach:** Reseachers can explore polynomial bases and the matrix polynomial function from other perspectives:
> (1) Consider LinearGNN convergence properties as discussed in [this paper](https://proceedings.mlr.press/v139/xu21k/xu21k.pdf)
> (2) Investigate stability of $\\{g_k(L)x\\}_k$ basis vectors
>
> > **Q3-1:** Should we abandon spectral filtering approaches?
>
> 1. We don't think they should be abandoned altogether. As mentioned in Sec 3.4, the semantic meaning of U is good for some graphs. Additionally, Belkin & Niyogi (2008) attempted to provide a theoretical foundation showing that Laplacian-based methods on discrete manifolds can be feasible in certain cases.
>
> 2. Another possible exploration without abandoning GFB is to **transform general graphs into subgraphs** where their eigenvectors possess semantic meaning, e.g., [this thesis](https://www2.eecs.berkeley.edu/Pubs/TechRpts/2013/EECS-2013-209.pdf).
>
> > **Q3-2:** Could alternative graph dictionaries provide a better foundation? What would be a good starting point?
>
> Alternative graph dictionaries are at least worth considering. At minimum, SGNN researchers should return to their initial intuition and find what beyond GFB have been explored by former scholars. For example, Shuman et al. (2013) proposed windowed GFT, but this method has high complexity on larger graphs. Here, perhaps combining it with coarsening methods for hierarchical graph processing could be beneficial.
>
> ---
>
> **II. Response to Weaknesses**
>
> Your main criticism is that we haven't proposed concrete alternative solutions, limiting practical impact.
>
> However, position track papers are not required to propose specific alternatives. The CFP states that position papers can "**generally adopt a meta-level perspective**", and our work aligns with this requirement by providing a deep, systematic analysis of how  Concern 1 has become a "self-evident truth" in the field. We trace its historical roots, examine its mathematical foundations.
> However, through our discussions in the questions section, we believe it would be valuable to propose some open-ended potential directions in our position paper.
>
> PS: The CFP requires an **Alternative view** section to articulate opposing views, not specific solutions. Since our opposing view is considered "self-evident," we kept this section brief but extensively analyzed its formation mechanism in preceding sections.
>
> ---
>
> Best regards,
> the authors

---

### Official Review · Reviewer_ctYz · 2025-03-14

**Significance:** 2
**Argument Clarity:** 3
**Rating:** 3
**Confidence:** 3

**Questions:**

1. In Section 4.6, the authors mention the trade-off between expressiveness and generalizability as a key concern in spectral GNNs. Could the authors provide concrete examples or scenarios where this trade-off becomes a more pressing issue in spectral GNNs, especially in contrast to classical Fourier analysis?
2. In Section 3.4, the authors argue that low-frequency eigenvectors are widely used in practice, and that spectral graphs often work well in practical domains. If that is the case, why is the issue with GFBs problematic in real-world scenarios?  Graphs with well-behaved eigenvectors do not seem to be a "special" case from a practical perspective, otherwise issues related to the limitations of GFB would have already manifested more prominently. Could the authors provide examples where GFB directly leads to performance degradation or instability in practical tasks?
3. Regarding Theorems 4.2, 4.4, and 4.6, it makes sense that having small $C$, $K$, and $\alpha$ leads to small upper bounds and better stability, the converse is not necessarily true. Could there be cases where the system remains stable despite the theoretical upper bound being large? When and why do such upper bounds become tight, thereby harming stability and generalization of spectral GNNs?

**Discussion Potential:**

3

**Paper Summary:**

This position paper critically examines the foundations of spectral GNNs, questioning the common assumption that graph Laplacian eigenvectors serve as a Fourier basis of the given graph, inheriting similar semantics as the continuous Fourier basis. The authors argue that due to the irregular and discrete nature of graph structures, such graph Fourier basis often fail to exhibit the expected global frequency behavior, especially in high-frequency components where localization effects dominate. Furthermore, the paper highlights the fundamental limitations in the higher-order filtering polynomial, demonstrating that complex filters inherently undermine stability and generalization. To address these concerns, the authors advocate for re-examining the working mechanisms of spectral GNNs.

**Position:**

Yes

**Position In Title:**

Yes

**Related Work:**

3

**Strengths And Weaknesses:**

### **Strengths**
1. The paper is well-written, with a clear and focused argument throughout.
2. The paper exhibits a thorough understanding of existing literature with a well-integrated survey of prior work.
3. The authors effectively identify and explain the potential pitfalls of graph Fourier basis in Section 3. They provide convincing empirical evidence with mesh graphs or Cora dataset for Concern 1.
4. The paper encourages the community to rethink the foundations of spectral GNNs, which could inspire further discussion and empirical work.

### **Weaknesses**
1. It is unclear whether the raised concerns are novel contributions. Similar issues have been discussed in prior literature [1-3] and the paper could have done more to distinguish its unique perspective.
2. As mentioned Section 4.6, the trade-off between expressiveness and generalizability is prevalent in general approximation theory, yet the paper lacks a sufficient theoretical or empirical justification for why it poses a more significant issue in spectral GNNs.
3. Unlike Concern 1 (Section 3), Concern 2 (Section 4) is not strongly supported by empirical evidence. There are no experiments demonstrating how increasing the polynomial order or filter complexity impacts stability or generalization in practice, especially in spectral GNNs compared to classical domains.
4. Key experimental results such as Appendix E are relegated to the appendix. Reducing the length of less critical discussions (e.g., Section 3.4) and moving empirical evidences that help convincing researchers into the main text would improve the paper's clarity and impact.

### **References**
1. Shuman et al., Vertex-frequency analysis on graphs. Applied and Computational Harmonic Analysis, 40(2), 260-291, 2016.
2. Levie et al.,  CayleyNets: Graph convolutional neural networks with complex rational spectral filters. IEEE Transactions on Signal Processing, 67(1), 97-109, 2018.
3. Bianchi et al., Graph neural networks with convolutional ARMA filters. IEEE Transactions on Pattern Analysis and Machine Intelligence, 44(7), 3496-3507, 2021.

**Support:**

2

---

> ### Author Rebuttal · Authors · 2025-04-01
>
> Dear Reviewer:
>
> Thank you for your suggestions regarding our work. You have raised several important questions and suggestions:
>
> I. Our special aspects
> - I-A. Concern 1: Since Shuman et al. (2013), Levie et al. (CayleyNets, 2018), and Bianchi et al. (ARMA, 2021) have also pointed out issues with high-frequency bases, what makes our Concern 1 in the Position paper special? (W1)
> - I-B. Concern 2: Since the tension between expressiveness and generalization exists widely, why is it particularly important in spectral GNNs? (W2, Q1)
>
> II. Concerning Concern 2, the stability and generalization issues need clearer explanation, including:
> - II-A. Empirical validation (Q2, W2)
> - II-B. More discussion of several upper bounds (Q3)
>
> ---
>
> **I-A. Special aspects of Concern 1**
>
> In Concern 1, we focus on whether the "graph Fourier basis" truly serves the semantic role of a Fourier basis. We noticed that researchers in the Spectral GNN field almost take this for granted, so we further questioned: **"Why has such a questionable belief been considered self-evident?"**
>
> Why do we think this question is important? **CayleyNet** was proposed 7 years ago, and the works listed in Section 2.1 (Background - Polynomial filters) almost all came after CayleyNet. It is evident that the questionable semantic nature of graph Fourier basis, as noted by CayleyNet, still holds great appeal for subsequent GNN researchers. Such a widely followed belief must have its intrinsic reasons, which we attempt to address in Section 3.4.
>
> Moreover, our examination of "graph Fourier basis may not be a reasonable dictionary" is more careful.
>
> Let us specifically compare our work with the ones you mentioned:
>
> - **Versus CayleyNet.** CayleyNet insightfully questioned the rationality of the high-frequency part U and proposed a compressed spectrum alternative. However, our reflection is stronger, deeper, and more comprehensive.
> - **Versus Shuman et al. (2013).** Our contribution is not mutually exclusive with such GSP work, as SGNN researchers adopted GSP's mathematical tools while neglecting the concerns that GSP had already raised.
> - **Versus ARMA.** ARMA provides a way to utilize graph Fourier basis different from polynomial filters, but still aims to use graph Fourier basis without pointing out potential issues in U.
>
> **I-B. Special aspects of Concern 2**
>
> The reason why the compromise in expressiveness and stability/generalization is particularly important in SGNN is mainly because its expressiveness directly corresponds to "each Fourier atom are linear independent to express any target signal".
>
> As in the signal and convolution perspective, these atoms should be "separated." In polynomial filters, since different frequency components are actually separated by large Lipschitz numbers, a mild h means that when modulating one Fourier atom, the response of nearby atoms will also change to a similar magnitude, which **differs significantly from the motivation from GSP**.
>
> ---
>
> **II-A. Empirical Evidence for Concern 2**
>
> Due to the time constraints of the rebuttal, we have added a simple stability experiment on cora, which you can see here: https://anonymous.4open.science/r/Images266-7F7E/exp.png. It can be seen that the Lipschitz constant of h(L) is related to graph structure sensitivity, while ||h(L)|| is related to feature sensitivity.
>
> **II-B. Further Explanation of Upper Bounds in Theorems**
>
> This issue is indeed important. We believe that the "complexity" and "response magnitude" of polynomials influence stability and generalization, and they are constrained by C, K, and α, but obviously, when K is large, polynomials can
> still be simple and have small norm values. Thus, the bound in Thm 4.4 is indeed too loose, as we  can see from the fact that it even skips  consideration of h's shape. The consideration of C and α is more subtle: suppose h has a place where C is large, but there is no projection of x in the eigencomponents near this position, or the eigenvectors corresponding to this position are not affected by graph structure perturbations, then even if the filter is complex, it won't cause stability issues in Thm 4.2.
>
> For concrete cases,
> - For Thm 4.2 let us consider the following case: assume that $i^* = {\rm argmin} _i h'(\lambda _i) $ and
> $\tilde{\mathbf{x}}$ be the one-hot vector where only the $i^*$-th coordinate equals one, the upper bound become $\Vert h _{\alpha}(\hat{\mathbf{L}}') - h _{\alpha }(\hat{\mathbf{L}}) \Vert \leq \varepsilon (\delta + 1) h'(\lambda _{i^*}) + \mathcal{O} (\varepsilon^2)$.
> Under this case, we think that the upper bound is nearly tight, since the inequality comes only from triangle inequality.
>
> - For Thm 4.4, the author employed various techniques for estimation in the proof to bound the maximum modulus of complex polynomials on a disk. Therefore, it is difficult to find specific cases where equality holds at every step (and thus achieve the tight upper bound).
>
> ---
>
> Best regards,
>
> the authors

---

> > ### Comment · Reviewer_ctYz · 2025-04-08
> >
> > We thank the authors for their detailed and thoughtful rebuttal. Most of the concerns are convincingly addressed.
> >
> > However, I still have a question about Section II-B. In the example for Theorem 4.2, shouldn't it be an argmax rather than argmin, since the Lipschitz constant involves a supremum? Also, I would appreciate further elaboration on the example itself, as I’m having trouble understanding how the bound becomes tight.
> >
> > In this regard, I will keep my score unchanged for now.

---

> > > ### Author Response · Authors · 2025-04-09
> > >
> > > Dear Reviewer,
> > >
> > > Thank you very much for your response to further discuss this matter. Through this, we have gained a deeper understanding of the theorem's proof.
> > >
> > > *(Note: For convenience of explanation, we directly use the notation from the original paper \url{https://arxiv.org/pdf/1905.04497}.)*
> > >
> > > ---
> > >
> > > **I. Overview of the Original Paper’s Proof**
> > >
> > > Let me first present to you the proof flow from the original paper.
> > > Please check it here: https://anonymous.4open.science/r/Images266-7F7E/thm_4_2_proof_illus.jpg .
> > > We illustrate how the proof **separate** the perturbed term into three parts with a diagram.
> > > Our example closely follows the second part.
> > >
> > >
> > > **II. Additional Explanation of the Example**
> > >
> > > Now, let's focus on equation (62) (the original paper has an equals sign here, which appears to be a typo):
> > >
> > > $$
> > > \left\|\sum\_{i=1}^N \tilde{x}\_i m\_i h'(\lambda\_i)\mathbf{v}\_i\right\|^2
> > > \leq \sum\_{i=1}^N
> > > \bigl|\tilde{x}\_i\bigr|^2
> > > \bigl|m\_i\bigr|^2
> > > \bigl|h'(\lambda\_i)\bigr|^2
> > > \|\mathbf{v}\_i\|^2.
> > > $$
> > >
> > > Here,
> > >
> > > - $\mathbf{v}\_i$: the $i$-th eigenvector of the original propagation matrix $\mathbf{S}$ (unperturbed);
> > > - $\hat{x}\_i$: the component of input signal $\mathbf{x}$ on $\mathbf{v}\_i$.
> > > - $m\_i$: the $i$-th eigenvector of the perturbation matrix $\mathbf{E}$; it participates in the separation of the two parts shown in the diagram above through Lemma 1.
> > >
> > > Since $\\\{ \mathbf{v}\_i \\\}\_{i=1}^{N}$ are mutually orthogonal, the above inequality achieves equality when we let $\mathbf{x}$’s energy is **concentrated** on a single $\mathbf{v}\_{i^{\star}}$ ,
> > > i.e., $\mathbf{x} = \mathbf{v}\_{i^{\star}}$.  In this case, the term
> > >
> > > $$
> > > \left\|\sum\_{i=1}^N \tilde{x}\_i m\_i h'(\lambda\_i) \mathbf{v}\_i\right\|^2
> > > $$
> > >
> > > becomes
> > >
> > > $$
> > > \|\hat{x}\_{i^{\star}} m\_{i^{*}} h'(\lambda\_{i^{\star}}) \mathbf{v}\_{i^{\star}}\|^2.
> > > $$
> > >
> > > Noting that $\hat{x}\_{i^{\star}} = \|\mathbf{x}\|$ and $\|\mathbf{v}\_{i^{\star}}\|=1$ at this point, this term further becomes
> > > $$
> > > \| m\_{i^{\star}} h'(\lambda\_{i^{\star}}) \| \|\mathbf{x}\|.
> > > $$
> > >
> > > At this point, we let $m\_{i^\star}$ correspond to the largest eigenvalue of $\mathbf{E}$, and let $h$ reach its maximum Lipschitz number at $\lambda\_{i^{\star}}$, then this term exactly equals $\epsilon C \|\mathbf{x}\|$.
> > >
> > > Now, let's look at equation (66). From equation (65) to (66), another triangle inequality is used, as shown below equation (66) in the original paper:
> > >
> > > $$
> > > \sum\_{i=1}^{N} \|\hat{x}\_i\| \leq \sqrt{N} \|\hat{\mathbf{x}}\| = \sqrt{N} \|{\mathbf{x}}\|.
> > > $$
> > >
> > > Note that we previously set the response vector $\hat{\mathbf{x}}$ to be one-hot (i.e., $\mathbf{x} := \mathbf{v}\_{i^{*}}$), so in our example, $\sqrt{N}$ is discarded, and this term is just $\|\mathbf{x}\|$, not $\sqrt{N}\|\mathbf{x}\|$.
> > >
> > > Therefore, combining these, the equation becomes (Here, we use our notation again, instead the 2019 paper's, to clarify the previous rebuttal)
> > > $$
> > > \| h\_{\alpha}(\hat{L}') - h\_{\alpha}(\hat{L}) \|
> > > \le
> > > \varepsilon(\delta + 1) h'(\lambda\_{i^*}) + \mathcal{O}\bigl(\varepsilon^2\bigr).
> > > $$
> > >
> > > **A More Intuitive Example.**
> > >
> > > We can consider a more intuitive example, which is essentially the same as our previous example except for one point.
> > >
> > > Let the propagation matrix be $\mathbf{S} = \mathbf{V} \mathbf{\Lambda} \mathbf{V}^{\mathrm{H}}$, and the perturbation matrix be $\mathbf{E} = \delta \mathbf{v}\_1 \mathbf{v}\_1^{\mathrm{H}}$. Clearly, this perturbation matrix only affects the response of the first eigenvalue $\mathbf{\lambda}\_1$, without affecting the responses of other eigenvalues or causing eigenvector misalignment effects.
> > > We similarly “require” the filter to reach its steepest point near $\lambda\_1$. Since misalignment is 0, in this case we get:
> > > $$
> > > \| h\_{\alpha}(\hat{L}') - h\_{\alpha}(\hat{L}) \|
> > > \le
> > > \varepsilon \delta h'(\lambda\_{1}) + \mathcal{O}\bigl(\varepsilon^2\bigr).
> > > $$
> > >
> > > This equation helps in understanding the perturbation analysis part that does not involve misalignment.
> > >
> > > **III. Further Discussion**
> > >
> > > We note that the misalignment term makes the problem much more complex.
> > >
> > > We attempted to improve the original paper's equation (66), for example, by trying to move the effect of $\mathbf{E}\_{U}$ to either the beginning or the end — this might have allowed us to remove the $\sqrt{N}$ term, but it proved difficult to achieve. Also, in the lecture notes of one of the authors (<https://gnn.seas.upenn.edu/wp-content/uploads/2020/10/lecture\_6\_handoutb.pdf>), he states: “Eigenvector misalignment is an **uncontrollable** property of the perturbation”.
> > >
> > > ---
> > >
> > > Thanks again for the in-depth discussion.
> > > I believe that in future papers, we should explain the theorem more intuitively.
> > > It would be good if we have the opportunity to improve our score.
> > >
> > > Best regards,
> > >
> > > Authors

---

### Decision · Program_Chairs · 2025-04-30

**Decision:**

Accept (poster)

**Comment:**

The paper argues that the current mainstream theoretical understanding of why spectral GNNs work is flawed, and calls for more research in this area. The reviewers agree that the paper provides sound theoretical and empirical arguments supporting their position.

The reviewers also point out to two presumed weaknesses of the paper, which in my view are not weaknesses for a paper in the position paper track. The first presumed weakness is that the paper does not provide an alternative theoretical explanation to replace the criticised one – but this is exactly the goal of the position paper track: to allow for papers that call for research in a certain direction without yet knowing where this research is going to lead. The second weakness is that the paper has limited practical implications – which is true, but at the same time a completely understandable for a paper that calls for better *theoretical* understanding of a topic. While such understanding may lead to practical applications, it is hard to speculate about them before the understanding is established.

The topic is popular, argumentation in the paper interesting and sound, while the presumed weakness do not apply to this type of paper.

Note: the running title is "Submission and Formatting Instructions for ICML 2025" which should be changed to this paper's title.